# DIRECTION MATTERS: ON THE IMPLICIT BIAS OF STOCHASTIC GRADIENT DESCENT WITH MODERATE LEARNING RATE

**Jingfeng Wu**
Computer Science Department
Johns Hopkins University
Baltimore, MD 21218, USA
`uuujf@jhu.edu`

**Difan Zou**
Computer Science Department
University of California, Los Angeles
Los Angeles, CA 90095, USA
`knowzou@cs.ucla.edu`

**Vladimir Braverman**
Computer Science Department
Johns Hopkins University
Baltimore, MD 21218, USA
`vova@cs.jhu.edu`

**Quanquan Gu**
Computer Science Department
University of California, Los Angeles
Los Angeles, CA 90095, USA
`qgu@cs.ucla.edu`

## ABSTRACT

Understanding the algorithmic bias of *stochastic gradient descent* (SGD) is one of the key challenges in modern machine learning and deep learning theory. Most of the existing works, however, focus on *very small or even infinitesimal* learning rate regime, and fail to cover practical scenarios where the learning rate is *moderate and annealing*. In this paper, we make an initial attempt to characterize the particular regularization effect of SGD in the moderate learning rate regime by studying its behavior for optimizing an overparameterized linear regression problem. In this case, SGD and GD are known to converge to the unique minimum-norm solution; however, with the moderate and annealing learning rate, we show that they exhibit different *directional bias*: SGD converges along the large eigenvalue directions of the data matrix, while GD goes after the small eigenvalue directions. Furthermore, we show that such directional bias does matter when early stopping is adopted, where the SGD output is nearly optimal but the GD output is suboptimal. Finally, our theory explains several folk arts in practice used for SGD hyperparameter tuning, such as (1) linearly scaling the initial learning rate with batch size; and (2) overrunning SGD with high learning rate even when the loss stops decreasing.

## 1 INTRODUCTION

*Stochastic gradient descent* (SGD) and its variants play a key role in training deep learning models. From the optimization perspective, SGD is favorable in many aspects, e.g., scalability for large-scale models (He et al., 2016), parallelizability with big training data (Goyal et al., 2017), and rich theory for its convergence (Ghadimi & Lan, 2013; Gower et al., 2019). From the learning perspective, more surprisingly, overparameterized deep nets trained by SGD usually generalize well, even in the absence of explicit regularizers (Zhang et al., 2016; Keskar et al., 2016). This suggests that SGD favors certain "good" solutions among the numerous global optima of the overparameterized model. Such phenomenon is attributed to the *implicit bias* of SGD. It remains one of the key theoretical challenges to characterize the algorithmic bias of SGD, especially with moderate and annealing learning rate as typically used in practice (He et al., 2016; Keskar et al., 2016).

In the *small learning rate regime*, the regularization effect of SGD is relatively well understood, thanks to the recent advances on the implicit bias of *gradient descent* (GD) (Gunasekar et al., 2017; 2018a;b; Soudry et al., 2018; Ma et al., 2018; Li et al., 2018; Ji & Telgarsky, 2019b;a; Ji et al., 2020; Nacson et al., 2019a; Ali et al., 2019; Arora et al., 2019; Moroshko et al., 2020; Chizat &

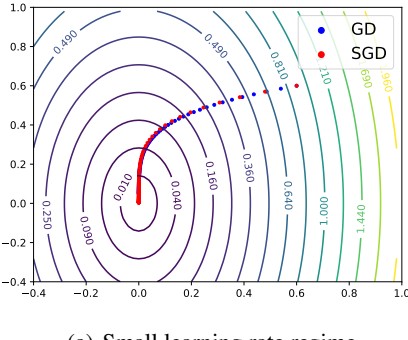
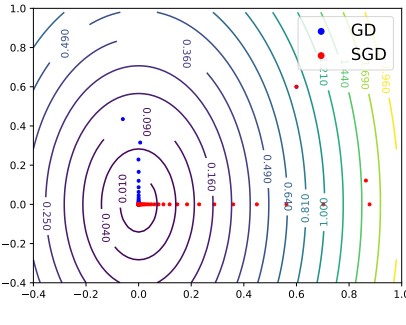

(a) Small learning rate regime

(b) Moderate learning rate regime

Figure 1: Illustration for the 2-D example studied in Section 3. Here $\kappa = 4$ and $w_0 = (0.6, 0.6)$. **(a)**: Small learning rate regime. The small learning rate is $0.1/\kappa$. In this regime SGD and GD behave similarly and they both converge along $e_2$. **(b)**: Moderate learning rate regime. The initial moderate learning rate is $\eta = 1.1/\kappa$ and the decayed learning rate is $\eta' = 0.1/\kappa$. In this regime GD converges along $e_2$ but SGD converges along $e_1$, the larger eigenvalue direction of the data matrix. Please refer to Section 3 for further discussions.

Bach, 2020). According to classical stochastic approximation theory (Kushner & Yin, 2003), with a sufficiently small learning rate, the randomness in SGD is negligible (which scales with learning rate), and as a consequence SGD will behave highly similar to its deterministic counterpart, i.e., GD. Based on this fact, the regularization effect of SGD with small learning rate can be understood through that of GD. Take linear models for example, GD has been shown to be biased towards max-margin/minimum-norm solutions depending on the problem setups (Soudry et al., 2018; Gunasekar et al., 2018a; Ali et al., 2019); correspondingly, follow-ups show that SGD with small learning rate has the same bias (up to certain small uncertainty governed by the learning rate) (Nacson et al., 2019b; Gunasekar et al., 2018a; Ali et al., 2020). The analogy between SGD and GD in the small learning rate regime is also demonstrated in Figures 1(a) and 3.

However, the regularization theory for SGD with small learning rate cannot explain the benefits of SGD in the *moderate learning rate regime*, where the initial learning rate is moderate and followed by annealing (Li et al., 2019; Nakkiran, 2020; Leclerc & Madry, 2020; Jastrzebski et al., 2019). In particular, empirical studies show that, in the moderate learning rate regime, (small batch) SGD generalizes much better than GD/large batch SGD (Keskar et al., 2016; Jastrzębski et al., 2017; Zhu et al., 2019; Wu et al., 2020) (see Figure 3). This observation implies that, instead of imitating the bias of GD as in the small learning rate regime, SGD in the moderate learning rate regime admits superior bias than GD — it requires a dedicated characterization for the implicit regularization effect of SGD with moderate learning rate.

In this paper, we reveal a particular regularization effect of SGD with moderate learning rate that involves *convergence direction*. In specific, we consider an overparameterized linear regression model learned by SGD/GD. In this setting, SGD and GD are known to converge to the unique minimum-norm solution (Zhang et al., 2016; Gunasekar et al., 2018a) (see also Section 2.1). However, with a moderate and annealing learning rate, we show that SGD and GD favor different convergence directions: *SGD converges along the large eigenvalue directions of the data matrix*; in contrast, GD goes after the small eigenvalue directions. The phenomenon is illustrated in Figure 1(b). To sum up, we make the following contributions in this work:

1. For an overparameterized linear regression model, we show that SGD with moderate learning rate converges along the large eigenvalue directions of the data matrix, while GD goes after the small eigenvalue directions. To our knowledge, this result initiates the regularization theory for SGD in the moderate learning rate regime, and complements existing results for the small learning rate.
2. Furthermore, we show the particular directional bias of SGD with moderate learning rate benefits generalization when early stopping is used. This is because converging along the large eigenvalue directions (SGD) leads to nearly optimal solutions, while converging along the small eigenvalue directions (GD) can only give suboptimal solutions.
3. Finally, our results explain several folk arts for tuning SGD hyperparameters, such as (1) linearly scaling the initial learning rate with batch size (Goyal et al., 2017); and (2) overrunning SGD with high learning rate even when the loss stops decreasing (He et al., 2016).

## 2 PRELIMINARY

Let $(x, y) \in \mathbb{R}^d \times \mathbb{R}$ be a pair of $d$-dimensional feature vector and 1-dimensional label. We consider a linear regression problem with square loss defined as $\ell(x, y; w) := (w^\top x - y)^2$, where $w \in \mathbb{R}^d$ is the model parameter. Let $\mathcal{D}$ be the population distribution over $(x, y)$, then the test loss is $L_{\mathcal{D}}(w) := \mathbb{E}_{(x,y) \sim \mathcal{D}} [\ell(x, y; w)]$. Let $\mathcal{S} := \{(x_i, y_i)\}_{i=1}^n$ be a training set of $n$ data points drawn i.i.d. from the population distribution $\mathcal{D}$. Then the training/empirical loss is defined as the average of the individual loss over all training data points,

$$L_{\mathcal{S}}(w) := \frac{1}{n} \sum_{i=1}^n \ell_i(w), \quad \text{where } \ell_i(w) := \ell(x_i, y_i; w) = (w^\top x_i - y_i)^2.$$

We use $\{\eta_k\}$ to denote a *learning rate scheme* (LR). Then *gradient descent* (GD) iteratively performs the following update:

$$w_{k+1} = w_k - \eta_k \nabla L_{\mathcal{S}}(w_k) = w_k - \frac{2\eta_k}{n} \sum_{i=1}^n x_i(x_i^\top w_k - y_i). \tag{GD}$$

Next we introduce *mini-batch stochastic gradient descent* (SGD).[1] Let $b$ be the batch size. For simplicity suppose $n = mb$ for an integer $m$ (number of mini-batches). Then at each epoch, SGD first randomly partitions the training set into $m$ disjoint mini-batches with size $b$, and then sequentially performs $m$ updates using the stochastic gradients calculated over the $m$ mini-batches. Specifically, at the $k$-th epoch, let the mini-batch index sets be $\mathcal{B}_1^k, \mathcal{B}_2^k, \ldots, \mathcal{B}_m^k$, where $|\mathcal{B}_j^k| = b$ and $\bigcup_{j=1}^m \mathcal{B}_j^k = \{1, 2, \ldots, n\}$, then SGD takes $m$ updates as follows

$$w_{k,j+1} = w_{k,j} - \frac{\eta_k}{b} \sum_{i \in \mathcal{B}_j^k} \nabla \ell_i(w_{k,j}) = w_{k,j} - \frac{2\eta_k}{b} \sum_{i \in \mathcal{B}_j^k} x_i(x_i^\top w_{k,j} - y_i), \quad j = 1, \ldots, m. \tag{SGD}$$

We also write $w_{k+1} = w_{k,m+1}$ and $w_k = w_{k,1}$ to be consistent with notations in (GD).

### 2.1 THE MINIMUM-NORM BIAS

Before presenting our results on the directional bias, let us first recap the well-known minimum-norm bias for SGD/GD optimizing linear regression problem (Zhang et al., 2016; Gunasekar et al., 2018a; Belkin et al., 2019; Bartlett et al., 2020). We rewrite the training loss as $L_{\mathcal{S}}(w) = \frac{1}{n} \|X^\top w - Y\|_2^2$, where $X = (x_1, \ldots, x_n) \in \mathbb{R}^{d \times n}$ and $Y = (y_1, \ldots, y_n)^\top \in \mathbb{R}^n$. Then its global minima are given by $\mathcal{W}_* := \{w \in \mathbb{R}^d : Pw = w_*, w_* := X(X^\top X)^{-1} Y\}$, where $P$ is the projection operator onto the data manifold, i.e., the column space of $X$. We focus on overparameterized cases where $\mathcal{W}_*$ contains multiple elements.

Notice that every gradient $\nabla \ell_i(w) = 2x_i(x_i^\top w - y_i)$ is spanned in the data manifold, thus (GD) and (SGD) can never move along the direction that is orthogonal to the data manifold. In other words, (GD) and (SGD) implicitly admit the following *hypothesis class*:

$$\mathcal{H}_{\mathcal{S}} = \{w \in \mathbb{R}^d : P_\perp w = P_\perp w_0\}, \tag{1}$$

where $w_0$ is the initialization and $P_\perp = I - P$ is the projection operator onto the orthogonal complement to the column space of $X$.

Putting things together, for any global optimum $w \in \mathcal{W}_*$ (hence $Pw = w_*$), we have

$$\|w - w_0\|_2^2 = \|Pw - Pw_0\|_2^2 + \|P_\perp w - P_\perp w_0\|_2^2 = \|w_* - Pw_0\|_2^2 + \|P_\perp w - P_\perp w_0\|_2^2,$$

where the right hand side is minimized when $P_\perp w = P_\perp w_0$, i.e., $w \in \mathcal{H}_{\mathcal{S}}$, thus $w$ is the solution found by SGD/GD in the non-degenerated cases (when the learning rate is set properly so that the algorithms can find a global optimum). In sum, SGD/GD is biased to find the global optimum that is *closest to the initialization*, which is referred as the "minimum-norm" bias in literature since the initialization is usually set to be zero.

---

[1] In this paper we focus on SGD without replacement, nonetheless our results and techniques are ready to be extended to SGD with replacement as well.

## 3 WARMING UP: A 2-DIMENSIONAL CASE STUDY

In this section we conduct a 2-dimensional case study to motivate our understanding on the directional bias of SGD in the moderate learning rate regime. Let us consider a training set consisting of two orthogonal points, $\mathcal{S} = \{(x_1, \ y_1 = 0), \ (x_2, \ y_2 = 0)\}$ where

$$x_1 = \sqrt{\kappa} \cdot e_1 = (\sqrt{\kappa}, \ 0)^\top, \quad x_2 = e_2 = (0, \ 1)^\top, \quad \kappa > 2.$$

Clearly $w_* = 0$ is the unique minimum of $L_{\mathcal{S}}(w)$. The Hessian of the empirical loss is $\nabla^2 L_{\mathcal{S}}(w) = x_1 x_1^\top + x_2 x_2^\top = \mathrm{diag}\,(\kappa, 1)$, which has two eigenvalues: the smaller one $1$ is contributed by data $x_2$, and the larger one $\kappa$ contributed by data $x_1$. Hence $L_{\mathcal{S}}(w)$ is $\kappa$-smooth. Similarly the Hessian of the individual losses are $\nabla^2 \ell_1(w) = 2x_1 x_1^\top = \mathrm{diag}\,(2\kappa, 0)$ and $\nabla^2 \ell_2(w) = 2x_2 x_2^\top = \mathrm{diag}\,(0, 2)$. Thus $\ell_2(w)$ is 2-smooth, but $\ell_1(w)$, the individual loss for data $x_1$, is only $2\kappa$-smooth, which is more ill-conditioned compared to $L_{\mathcal{S}}(w)$ and $\ell_2(w)$.

Next we consider a moderate initial learning rate $\eta \in \left(\frac{1}{\kappa}, \ \frac{2}{1+\kappa}\right)$. According to convex optimization theory ([Boyd et al., 2004](#)), gradient step with such learning rate is convergent for $L_{\mathcal{S}}(w)$ and $\ell_2(w)$, but oscillating for $\ell_1(w)$. In other words, (GD) is convergent; and (SGD) is convergent along direction $x_2$ (or $e_2$), but oscillating along direction $x_1$ (or $e_1$). We also see this by analytically solving (GD) and (SGD) for this example:

$$w_k^{\mathrm{gd}} = \begin{pmatrix} (1 - \eta\kappa)^k & \\ & (1 - \eta)^k \end{pmatrix} w_0, \quad w_k^{\mathrm{sgd}} = \begin{pmatrix} (1 - 2\eta\kappa)^k & \\ & (1 - 2\eta)^k \end{pmatrix} w_0, \qquad (2)$$

where $|1 - \eta\kappa| < |1 - \eta| < 1$ and $|1 - 2\eta| < 1 < |1 - 2\eta\kappa|$.

By Eq. (2), with moderate learning rate GD is convergent for both directions $e_1$ and $e_2$. Moreover, GD fits $e_1$ faster since the contraction parameter is smaller, i.e., $|1 - \eta\kappa| < |1 - \eta| < 1$. Thus observing the entire optimization path, GD approaches the minimum $w_* = 0$ along $e_2$, which corresponds to the smaller eigenvalue direction of $\nabla^2 L_{\mathcal{S}}(w)$. This is verified by the blue dots in Figure 1(b). We note this directional bias for GD also holds in the small learning rate regime, as shown in Figure 1(a).

As for SGD in the initial phase where the learning rate is moderate, Eq. (2) shows it converges along $e_2$ but oscillates along $e_1$ since $|1 - 2\eta| < 1 < |1 - 2\eta\kappa|$. In other words, SGD cannot fit $e_1$ before the learning rate decays; however when this happens, $e_2$ is already well fitted. Overall, SGD fits $e_2$ first then fits $e_1$, i.e., SGD converges to the minimum $w_* = 0$ along $e_1$, which corresponds to the larger eigenvalue direction of $\nabla^2 L_{\mathcal{S}}(w)$. This is verified by the red dots in Figure 1(b). We note this particular directional bias for SGD is dedicated to the moderate learning rate regime; in the small learning rate regime, as discussed before, SGD behaves similar to GD thus goes after the smaller eigenvalue direction, which is illustrated in Figure 1(a).

The above idea can be carried over to more general cases: the training loss usually has relatively smooth curvature because of the empirical averaging; yet some individual losses can possess bad smoothness condition, corresponding to the data points that contribute to the large eigenvalues of the Hessian/data matrix. Then with a moderate learning rate, while GD is convergent, SGD is convergent for the smooth individual losses but oscillating for the ill-conditioned individual losses. Thus SGD can only fit the latter losses after the learning rate anneals. Therefore, in the moderate learning rate regime, SGD tends to converge along the large eigenvalue directions while GD tends to go after the small eigenvalue directions. We will rigorously justify the above intuitions in the following section.

## 4 MAIN RESULTS

In this section we present our main theoretical results. The proofs are deferred to Appendix B.

We specify the population distribution of $(x, y) \in \mathbb{R}^d \times \mathbb{R}$ in the following manner. (1) We consider the feature vector as $x = \zeta \cdot \xi$, where $\zeta$ and $\xi$ are two independent random variables that represent the magnitude and angle of $x$, respectively. That is, $\zeta \in \mathbb{R}$ is bounded in $(0, 1]$, and $\xi \in \mathbb{R}^d$ obeys a sphere uniform distribution, $\mathcal{U}(S^{d-1})$. (2) We consider a realizable setting where the label is given by $y = w_*^\top x$, i.e., there exists a true parameter $w_* \in \mathbb{R}^d$ that generates the label from the feature

vector[2]. Then the test loss is $L_{\mathcal{D}}(w) = \mathbb{E}_{(x,y)\sim\mathcal{D}} \left[ (w - w_*)^\top x x^\top (w - w_*) \right] = \mu \|w - w_*\|_2^2$, where $\mu = \mathbb{E}[\zeta^2]/d$. For an i.i.d. generated training set $\mathcal{S} = \{(x_i, y_i)\}_{i=1}^n$, the training loss and the individual losses are

$$L_{\mathcal{S}}(w) = \frac{1}{n} (w - w_*)^\top X X^\top (w - w_*), \quad \ell_i(w) = (w - w_*)^\top x_i x_i^\top (w - w_*), \quad i = 1, \ldots, n,$$

where $X = (x_1, \ldots, x_n)$. We denote by $P$ the projection operator onto the column space of $X$ (the data manifold). For $i \in [n]$, we denote $\lambda_i := \|x_i\|_2^2 = \zeta_i^2 \in (0, 1]$. Without loss of generality, we assume $\{\lambda_i\}_{i\in[n]}$ are sorted in a descending order, i.e., $\lambda_1 \geq \lambda_2 \geq \cdots \geq \lambda_n$. With the these preparations, we are ready to state our main theorems.

## 4.1 THE DIRECTIONAL BIAS OF SGD

We first present Theorems 1 and 2 that characterize the different directional biases of SGD and GD in the moderate learning rate regime.

**Theorem 1** (The directional bias of SGD with moderate LR, informal). *Suppose $d \geq \text{poly}(n)$[3]. Denote $\nu = n/\sqrt{d}$ (which is small). Then with high probability it holds that $\lambda_1 > \lambda_2 + \Theta(\nu)$, $\lambda_{n-1} > \lambda_n + \Theta(\nu)$, $\lambda_n > \Theta(\nu)$. Suppose the initialization is set such that $x_i^\top (w_0 - w_*) \neq 0$ for every $i \in [n]$[4]. Consider (SGD) with the following moderate learning rate scheme*

$$\eta_k = \begin{cases} \eta \in \left( \frac{b}{\lambda_1 - \Theta(\nu)}, \frac{b}{\lambda_2 + \Theta(\nu)} \right), & k = 1, \ldots, k_1; \\ \eta' \in \left( 0, \frac{b}{2\lambda_1} \right), & k = k_1 + 1, \ldots, k_2, \end{cases} \tag{3}$$

*then for $\epsilon$ such that $\text{poly}(\epsilon) > \nu$, there exist $k_1 = \mathcal{O}\left( \log \frac{1}{\epsilon} + k_2 \right)$ and $k_2 > 0$ such that with high probability the output of SGD $w^{\text{sgd}} := w_{k_2}$ satisfies*

$$(1 - \epsilon) \cdot \gamma_1 \leq \frac{\left( P(w^{\text{sgd}} - w_*) \right)^\top \cdot X X^\top \cdot P\left( w^{\text{sgd}} - w_* \right)}{\|P(w^{\text{sgd}} - w_*)\|_2^2} \leq \gamma_1, \tag{4}$$

*where $\gamma_1$ is the largest eigenvalue of the data matrix $X X^\top$.*

**Theorem 2** (The directional bias of GD with moderate or small LR, informal). *Under the same conditions as Theorem 1, consider (GD) with the following moderate or small learning rate scheme*

$$\eta_k \in \left( 0, \frac{n}{2\lambda_1 + \Theta(\nu)} \right), \quad k = 1, \ldots, k_2, \tag{5}$$

*then for any $\epsilon > 0$, if $k_2 > \mathcal{O}\left( \log \frac{1}{\epsilon} \right)$, then with high probability the output of GD $w^{\text{gd}} := w_{k_2}$ satisfies*

$$\gamma_n \leq \frac{\left( P(w^{\text{gd}} - w_*) \right)^\top \cdot X X^\top \cdot P\left( w^{\text{gd}} - w_* \right)}{\|P(w^{\text{gd}} - w_*)\|_2^2} \leq (1 + \epsilon) \cdot \gamma_n, \tag{6}$$

*where $\gamma_n$ is the smallest eigenvalue of the data matrix $X X^\top$ restricted in the column space of $X$.*

**Remark 1.** *As the Rayleigh quotient (4) (resp. (6)) converges to its maximum (resp. minimum), the vector gets closer to the eigenvector of the largest (resp. smallest) eigenvalue (Trefethen & Bau III, 1997). Thus Theorem 1 and 2 suggest that, when projected onto the data manifold, SGD and GD converge to the optimum along the largest and smallest eigenvalue direction respectively. Here we are only interested in the projection onto the data manifold, since SGD/GD cannot move along the direction that is orthogonal to the data manifold as discussed in Section 2.1.*

**Remark 2.** *In Theorem 1 we use the gap between $\lambda_1$ and $\lambda_2$ to show a learning rate scheme such that SGD converges along the largest eigenvalue direction. This can be extended by considering the gap between $\lambda_r$ and $\lambda_{r+1}$ and the learning rate scheme defined similarly, then SGD converges along the subspace spanned by the eigenvectors of the top $r$ eigenvalues. Similar extension applies to Theorem 2 as well.*

---

[2]This is for the conciseness of presentation. Our results can be easily generalized to linear regression with well-specified noise, i.e., noise that is independent of the feature vector.

[3]For two sequences $\{x_n \geq 0\}$ and $\{y_n \geq 0\}$: $x_n = \mathcal{O}(y_n)$ if there exist constants $C > 0$ and $N$ such that $x_n \leq C y_n$ for every $n \geq N$; $x_n = \Theta(y_n)$ if $x_n = \mathcal{O}(y_n)$ and $y_n = \mathcal{O}(x_n)$. $x_n = \text{o}(y_n)$ if for every $\epsilon > 0$ there exists a positive constant $N(\epsilon) > 0$ such that $x_n \leq C y_n$ for every $n \geq N(\epsilon)$; $x_n = \text{poly}(y_n)$ if there exists large absolute constant $D > 0$ such that $x_n = \Theta\left( y_n^D \right)$.

[4]This holds with probability 1 if $w_0$ is initialized randomly and follows, e.g., Gaussian distribution.

**Remark 3.** *We legitimately assume* $b < n/2 - \Theta(\nu)$ *since it is not very meaningful to discuss SGD that uses more than (roughly) half of the training set as a mini-batch. Then the learning rate schedule in* (3) *intersects with that in* (5)*, i.e.,* (5) *covers both moderate and small learning rate schemes. Their intersection determines a moderate learning rate scheme, where SGD converges along the large eigenvalue directions while GD goes after the small eigenvalue directions. This justifies the regularization effect of SGD with moderate learning rate.*

**Remark 4.** *Technically, in Theorem* 1 *one can set* $b = n$ *to include GD as a special case, so that GD also follows the large eigenvalue directions. However, the initial learning rate in* (3) *needs to be at least* $\frac{n}{\lambda_1} \geq n$ *ignoring the small order term, which is way too large to be even numerically stable in practical big data circumstances. This observation is also directly supported by Theorem* 2*, where* (5) *specifies the range of learning rate such that GD converges along small eigenvalue directions. Note the upper bound in* (5) *linearly scales with* $n$*, and is large enough to include all learning rate that can be adopted in practice. Thus one cannot have a legitimate learning rate for GD to converge along the large eigenvalue directions as SGD does with moderate learning rate.*

Note GD with small learning rate also converges along the small eigenvalue directions, since (5) covers the small learning rate scheme. In complement, the following Theorem 3 shows that in the small learning rate regime, SGD is imitating GD and converges along the small eigenvalue directions as well. Theorems 1, 2 and 3 together show that, converging along the large eigenvalue directions is a distinct regularization effect that is unique to *SGD* with *moderate learning rate*.

**Theorem 3** (The directional bias of SGD with small LR, informal). *Theorem* 2 *applies to* (SGD) *with the following small learning rate scheme*

$$\eta_k = \eta' \in \left(0, \ \frac{b}{2\lambda_1 + \Theta(\nu)}\right), \quad k = 1, \ldots, k_2. \tag{7}$$

**Experiment** Figure 2 shows two experiments for verifying Theorems 1, 2 and 3. The details of the experimental setup are deferred to Appendix D. In Figure 2(a), we run SGD and GD in both moderate and small learning rate regimes, and directly compare their Rayleigh quotients as defined in (4) and (6). We can see that the Rayleigh quotient reaches its maximum for SGD with moderate learning rate, and reaches its minimum for both GD and SGD with small learning rate, which verifies Theorems 1, 2 and 3. In Figure 2(b), we run the algorithms to optimize a neural network on a subset of the FashionMNIST dataset. Since the neural network is non-convex and have multiple local minima, we compare the *relative Rayleigh quotients*, i.e., the Rayleigh quotients of the convergence directions divided by the maximum absolute eigenvalue of the Hessian (see Appendix D.3). Figure 2(b) shows that SGD with moderate learning rate converges along relatively large eigenvalue directions while GD/SGD with small learning rate converges along relatively small eigenvalue directions. This distinguishes the directional bias of SGD and GD in the moderate learning rate regime and provides evidence from neural network training to support our theory.

## 4.2 EFFECTS OF THE DIRECTIONAL BIAS

Next we justify the benefit of the particular directional bias of SGD with moderate learning rate.

Recall the hypothesis class $\mathcal{H}_{\mathcal{S}}$ (Eq. (1)) for SGD and GD. Then for an algorithm output $w^{\mathrm{alg}}$, we have the following generalization error decomposition (Shalev-Shwartz & Ben-David, 2014),

$$L_{\mathcal{D}}(w^{\mathrm{alg}}) - \inf_w L_{\mathcal{D}}(w) = \underbrace{L_{\mathcal{D}}(w^{\mathrm{alg}}) - \inf_{w' \in \mathcal{H}_{\mathcal{S}}} L_{\mathcal{D}}(w')}_{\Delta(w^{\mathrm{alg}}), \text{ estimation error}} + \underbrace{\inf_{w' \in \mathcal{H}_{\mathcal{S}}} L_{\mathcal{D}}(w') - \inf_w L_{\mathcal{D}}(w)}_{\text{approximation error}}.$$

The *approximate error* is an intrinsic error determined by the hypothesis class, and is not improvable unless enlarging the hypothesis class. In contrast, the *estimation error* $\Delta(w^{\mathrm{alg}})$ is determined by the algorithm as well as its hyperparameters. Thus, in the following theorem, we use the estimation error to compare the generalization performance of the SGD and GD outputs in different learning rate regimes.

**Theorem 4** (Effects of the directional bias, informal). *Let* $\mathcal{W}_\alpha := \{w \in \mathcal{H}_{\mathcal{S}} : L_{\mathcal{S}}(w) = \alpha\}$ *be an* $\alpha$*-level set of the training loss* $L_{\mathcal{S}}(w)$*. Let* $\Delta_\alpha^* := \inf_{w \in \mathcal{W}_\alpha} \Delta(w)$ *be the minimum estimation error within the* $\alpha$*-level set* $\mathcal{W}_\alpha$*. Under the same conditions as Theorems* 1*-* 3 *and assuming that* $b < n/2 - \Theta(\nu)$*, then the following holds with high probability:*

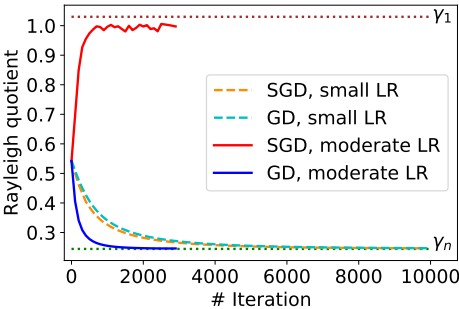 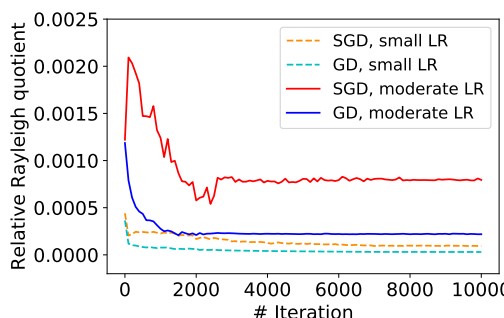

(a) Linear regression on synthetic data

(b) Neural network on a subset of FashionMNIST

Figure 2: Comparison of the (relative) Rayleigh quotients. **(a)**: A linear regression example. We randomly draw 100 samples from a $10,000$-dimensional space as described in Section 4, where $\zeta \sim \mathcal{U}([0.5, 1])$. The small learning rate scheme is specified by $(\eta', k_2) = (0.2, 10^4)$, and the moderate learning rate scheme is specified by $(\eta, \eta', k_1, k_2) = (1.05, 0.1, 2 \times 10^3, 3 \times 10^3)$. Numerical results show the Rayleigh quotient converges to its maximum for SGD with moderate learning rate, and converges to its minimum for GD and SGD with small learning rate, which verifies Theorems 1, 2 and 3. **(b)**: A neural network example. The plots are averaged over 10 runs. We randomly draw $2,000$ samples from FashionMNIST as the training set. The model is a 5-layer convolutional neural network. The small learning rate scheme is specified by $(\eta', k_2) = (10^{-3}, 10^4)$, and the moderate learning rate scheme is specified by $(\eta, \eta', k_1, k_2) = (10^{-2}, 10^{-3}, 2.5 \times 10^3, 10^4)$. Since neural network is non-convex, we compare the *relative Rayleigh quotient* of the concerned algorithms, i.e., the Rayleigh quotient of the convergence directions divided by the maximum absolute eigenvalue of the Hessian (see Appendix D.3).

- *The output of* (SGD) *with moderate LR* (3) *in Theorem 1 satisfies* $\Delta(w^{\mathrm{sgd}}) < (1 + \epsilon) \cdot \Delta_\alpha^*$, *where $\alpha$ is the training loss of $w^{\mathrm{sgd}}$ and $\epsilon$ is a small constant;*
- *The output of* (GD) *with moderate or small LR* (5) *in Theorem 2 satisfies* $\Delta(w^{\mathrm{gd}}) > M \cdot \Delta_\alpha^*$, *where $\alpha$ is the training loss of $w^{\mathrm{gd}}$ and $M = (1 - \epsilon) \cdot \gamma_1/\gamma_n$ is a constant larger than $1 + \epsilon$;*
- *The output of* (SGD) *with small LR* (7) *in Theorem 3 satisfies* $\Delta(w^{\mathrm{sgd}}) > M \cdot \Delta_\alpha^*$, *where $\alpha$ is the training loss of $w^{\mathrm{sgd}}$ and $M = (1 - \epsilon) \cdot \gamma_1/\gamma_n$ is a constant larger than $1 + \epsilon$.*

Theorem 4 suggests that: (1) in the moderate learning rate regime, there is a separation between the test error of SGD and that of GD. In detail, early stopped SGD finds a nearly optimal solution thanks to its particular directional bias. In contrast, early stopped GD can only find a suboptimal one; and (2) in the small learning rate regime, however, SGD no longer admits the dedicated directional bias for moderate learning rate. Instead it behaves similarly as GD, and hence outputs suboptimal solutions when early stopping is adopted.

**Remark 5.** *In practice it is usually intractable and unnecessary to achieve the exact global minima of the training loss; instead we often* early stop *the algorithm once obtaining a small enough training loss, i.e., reaching an $\alpha$-level set. In this spirit, Theorem 4 compares the generalization ability of SGD with moderate learning rate vs. GD/SGD with small learning rate within a level set.*

**Remark 6.** *We note the second conclusion in Theorem 4 is also obtained by Nakkiran (2020) for a different purpose. In specific, Nakkiran (2020) show the separation between the test error of GD with "large" and annealing learning rate, and test error of GD with small learning rate. However, the "large" learning rate for GD in their analysis is linear in the training sample size and is not practical as we have discussed in Remark 4. In contrast, we show that under the practically used moderate learning rate, there is a separation between the generalization abilities of SGD and GD. To our knowledge, our work gives the first theoretical justification of the phenomenon that SGD outperforms GD when the learning rate is moderate.*

**Experiment**   Figure 3 shows the generalization performance of a neural network trained by SGD and GD in both moderate and small learning rate regimes. The setup details are included in Appendix D. We can observe that (1) SGD with moderate learning rate generalizes the best, and (2) GD and SGD with small learning rate perform similarly, but both are worse than SGD with moderate learning rate. The empirical results suggest SGD with moderate learning has certain benign regularization effect. This is explained by the distinct directional bias for SGD with moderate learning rate shown in previous theorems.

## 5 RELATED WORK

In this section, we review related works and discuss their similarities and differences to our work.

**The implicit bias of GD**  The implicit bias of GD has been extensively studied in recent years. We summarize several representative results as follows. For homogeneous model with exponentially-tailed loss, GD converges along the max-margin direction (Gunasekar et al., 2017; 2018a;b; Soudry et al., 2018; Ma et al., 2018; Ji & Telgarsky, 2019a; Ji et al., 2020; Nacson et al., 2019a). For least square problem and its variants, GD is biased towards minimum-norm solution (Gunasekar et al., 2018a; Ali et al., 2019; Suggala et al., 2018). Note that this is also the foundation for the learning theory in the interpolation regime such as *double descent* (Belkin et al., 2019) and *benign overfitting* (Bartlett et al., 2020). For matrix factorization and linear network, Gunasekar et al. (2017); Li et al. (2018) show the GD solution minimizes nuclear norm in special cases; more generally, Arora et al. (2019); Ji & Telgarsky (2019b) show GD balances/aligns layers; Moroshko et al. (2020) suggest the GD bias relies on initialization scale and training accuracy. For infinite-width network, Chizat & Bach (2020) show GD finds a max-

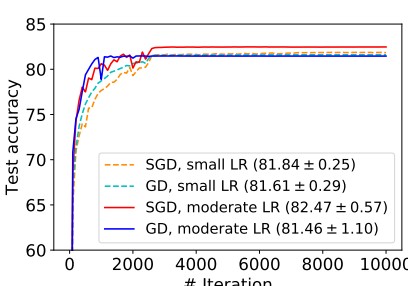

Figure 3: The test accuracy of a neural network on a subset of FashionMNIST. The plots are averaged over 10 runs. The experimental setting is identical to that in Figure 2(b). The plots show that SGD with moderate learning rate achieves the highest test accuracy, and GD and SGD with small learning rate perform similarly, but are worse than the former.

margin classifier in a functional space. The regularization theory for GD is fruitful; While some of them can be applied to SGD with small learning rate as have discussed (Nacson et al., 2019b; Gunasekar et al., 2018a; Ali et al., 2020), none of them can be carried over to SGD in the moderate learning rate regime. As far as we know, our paper is the first to study the regularization effect of SGD with moderate learning rate.

**The stability of SGD**  *Stability* is another approach to justify the generalization ability of SGD, where a stable algorithm is guaranteed to have small generalization error (Bousquet & Elisseeff, 2002). Along this line, several works show that SGD is stable under certain assumptions and therefore generalizes well (Hardt et al., 2016; Kuzborskij & Lampert, 2018; Charles & Papailiopoulos, 2018; Bassily et al., 2020). More interestingly, Charles & Papailiopoulos (2018) show a simple example where SGD is stable but GD is not, which partly explains the empirically superior performance of SGD. We also aim to justify the benefits of SGD, but take a different approach from the algorithmic regularization.

**The escaping behavior of SGD**  A popular theory (Jastrzębski et al., 2017; Zhu et al., 2019; Simsekli et al., 2019) attributes the regularization effect of SGD to its behavior of escaping from sharp minima. These works are built upon the continuous approximation of SGD via stochastic differential equations (Li et al., 2017; Hu et al., 2017; Xu et al., 2018; Simsekli et al., 2019). However it requires a small learning rate for the approximation to hold. In contrast, our main interest is in the moderate learning rate regime. Another related work in this line is (Wu et al., 2018), where they study the dynamical stability of minima and how SGD chooses them, but they do not show a directional bias introduced in our work.

**Non-small learning rate**  The regularization effect of non-small learning rate has received increasing attentions recently. Theoretically, Li et al. (2019); HaoChen et al. (2020) study the generalization of certain stochastic dynamics equipped with annealing learning rate scheme. However, their results cannot cover vanilla SGD. Lewkowycz et al. (2020) study the role of initial large learning rate for infinity-width network. Our work is motivated by Nakkiran (2020), which shows that a large initial learning rate helps GD generalize. This result can be recovered from our theorems; more importantly, we characterize the directional bias of SGD. Empirically, Jastrzebski et al. (2019); Leclerc & Madry (2020) investigate the impact of two-phase learning rate for training with SGD, but they do not provide any theoretical analysis.

## 6    DISCUSSIONS

**Linear scaling rule**  Linearly enlarging the initial learning rate according to batch size, or the *linear scaling rule* (Goyal et al., 2017), is an important folk art for paralleling SGD with large batch size while maintaining a good generalization performance. Interestingly, the linear scaling rule arises naturally from Theorem 1, where LR (3) suggests the initial learning rate $\eta$ to scale linearly with batch size $b$ to guarantee the desired directional bias. Our theory thus partly explains the mechanism of the linear scaling rule.

**Overrunning SGD with high learning rate**  Besides the moderate initial learning rate, another key ingredient in Theorem 1 is a sufficiently large $k_1$, i.e., SGD needs to run with moderate learning rate for sufficiently long time (to fit small eigenvalue directions). This requirement coincidentally agrees with the folk art to *overrun SGD with high learning rate*. For example, see Figure 4 in (He et al., 2016): from the $\sim 2 \times 10^5$-th to the $\sim 3 \times 10^5$-th iteration, even the training error seems to make no progress, practitioners let SGD run with a relatively high learning rate for nearly $10^5$ iterates. Our theory sheds light on understanding the hidden benefits in this "overrunning" phase: indeed the loss is not decreasing since SGD with high learning rate cannot fit the large eigenvalue directions, but on the other hand the overrunning lets SGD fit the small eigenvalue directions better, which in the end leads to the directional bias that SGD converges along the large eigenvalue directions according to Theorem 1. Thus overrunning SGD with high learning rate is helpful.

**Key technical challenges**  With non-small learning rate, analyzing SGD is usually hard since measure concentration turns vacuous and as a result one cannot relate the SGD iterates to that of GD. Alternatively, we control the SGD iterates epoch by epoch, then bound their composition to characterize the long run behavior of SGD. However, controlling the epoch-wise update of SGD is highly non-trivial, since in different epochs the sequence of stochastic gradient steps varies, and they do not commute due to the issue of matrix multiplication. To overcome this difficulty, we adopt techniques from matrix perturbation theory (Horn & Johnson, 2012) and conduct an analysis in the overparameterized regime. We believe these techniques are of independent interest.

## 7    CONCLUSION AND FUTURE WORK

We characterize a distinct directional regularization effect of SGD with moderate learning rate, where SGD converges along the large eigenvalue directions of the data matrix. In contrast, neither GD nor SGD with small learning rate can achieve this effect. Moreover, we show this directional bias benefits generalization when early stopping is adopted. Finally, our theory explains several folk arts used in practice for SGD hyperparameter tuning.

As an initial attempt, our results are limited to overparameterized linear models, and we ignore other factors that may contribute to the good generalization of SGD for training neural networks, e.g., network structures and explicit regularization. It is left as a future work to extend our results to nonlinear/nonconvex models (with explicit regularization).

### ACKNOWLEDGEMENT

We would like to thank the anonymous reviewers for their helpful comments. QG is partially supported by the National Science Foundation CAREER Award 1906169, IIS-2008981 and Salesforce Deep Learning Research Award. DZ is supported by the Bloomberg Data Science Ph.D. Fellowship. VB is supported in part by NSF CAREER grant 1652257, ONR Award N00014-18-1-2364 and the Lifelong Learning Machines program from DARPA/MTO. JW is supported by ONR Award N00014-18-1-2364. The views and conclusions contained in this paper are those of the authors and should not be interpreted as representing any funding agencies.

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

## A  PRELIMINARIES

### A.1  ADDITIONAL NOTATIONS

We adopt the notations and settings in main text. In addition we make the following notations.

For a vector $x \in \mathbb{R}^d$, denote its direction as $\bar{x} := \frac{x}{\|x\|_2}$. For simplicity assume the training data $\{x_1, \ldots, x_n\}$ are linear independent. For training data $x_i$, $i \in [n]$, we denote $\lambda_i = \|x_i\|_2^2$, then by construction we have $\lambda_i \in (0, 1]$. Without loss of generality let $\lambda_1 \geq \cdots \geq \lambda_n$. We define

$$X := (x_1, \ldots, x_n) \in \mathbb{R}^{d \times n},$$
$$X_{-1} := (x_2, x_3, \ldots, x_n) \in \mathbb{R}^{d \times (n-1)}.$$

Then based on the above definitions, we define the following two projection operators

$$P = X(X^\top X)^{-1} X^\top,$$
$$P_\perp = I - P.$$

Clearly for any $v \in \mathbb{R}^d$, $Pv$ projects $v$ onto subspace span $\{x_1, \ldots, x_n\}$, while $P_\perp v$ projects $v$ onto the orthogonal complement of span $\{x_1, \ldots, x_n\}$. Furthermore we introduce two more projection operators

$$P_{-1} = X_{-1}(X_{-1}^\top X_{-1})^{-1} X_{-1}^\top,$$
$$P_1 = P - P_{-1} = I - P_\perp - P_{-1}.$$

For any $v \in \mathbb{R}^d$, $P_{-1}v$ projects $v$ onto subspace span $\{x_2, \ldots, x_n\}$, while $P_1 v$ projects $v$ into the orthogonal complement of span $\{x_2, \ldots, x_n\}$ with respect to span $\{x_1, \ldots, x_n\}$. In the following, we often write the column space of $P$, which refers to $\{Pv : v \in \mathbb{R}^d\}$, similarly for $P_{-1}$, $P_1$ and $P_\perp$ as well. Clearly the column space of $P$ is also span $\{x_1, \ldots, x_n\}$; the column space of $P_{-1}$ is also the data manifold span $\{x_2, \ldots, x_n\}$. We highlight that the total space $\mathbb{R}^d$ can be decomposed as the direct sum of the column space of $P_{-1}$, $P_1$ and $P_\perp$, i.e., $I = P_{-1} + P_1 + P_\perp$. By definition, it is easy to verify that

$$P_\perp X = 0,$$
$$PX = X,$$
$$P_1 X = (P_1 x_1, 0, \ldots, 0),$$
$$P_{-1} X = (P_{-1} x_1, x_2, \ldots, x_n).$$

Then we define the following matrices which will be repeatedly used in the subsequent proof.

$$H := XX^\top,$$
$$H_{-1} := (P_{-1}X)(P_{-1}X)^\top,$$
$$H_1 := (P_1 X)(P_1 X)^\top,$$
$$H_c := (P_{-1}x_1)(P_1 x_1)^\top + (P_1 x_1)(P_{-1} x_1)^\top.$$

Based on the above definitions, it is easy to show that

$$
\begin{aligned}
H &= (P_1 X + P_{-1}X)(P_1 X + P_{-1}X)^\top \\
&= (P_{-1}X)(P_{-1}X)^\top + (P_1 X)(P_1 X)^\top + (P_1 X)(P_{-1}X)^\top + (P_{-1}X)(P_1 X)^\top \\
&= H_{-1} + H_1 + H_c.
\end{aligned}
$$

### A.2  LEMMAS

We present the following theorems and lemmas as preparation for our analysis.

**Theorem** (Gershgorin circle theorem, restated for symmetric matrix). *Let $A \in \mathbb{R}^{n \times n}$ be a symmetric matrix. Let $A_{ij}$ be the entry in the $i$-th row and the $j$-th column. Let*

$$R_i(A) := \sum_{j \neq i} |A_{ij}|, \ i = 1, \ldots, n.$$

*Consider $n$ Gershgorin discs*

$$D_i(A) := \{z \in \mathbb{R}, |z - A_{ii}| \leq R_i(A)\}, \; i = 1, \ldots, n.$$

*The eigenvalues of $A$ are in the union of Gershgorin discs*

$$G(A) := \bigcup_{i=1}^{n} D_i(A).$$

*Furthermore, if the union of $k$ of the $n$ discs that comprise $G(A)$ forms a set $G_k(A)$ that is disjoint from the remaining $n - k$ discs, then $G_k(A)$ contains exactly $k$ eigenvalues of A, counted according to their algebraic multiplicities.*

*Proof.* See, e.g., Horn & Johnson (2012), Chap 6.1, Theorem 6.1.1. □

**Theorem** (Hoffman-Wielandt theorem, restated for symmetric matrix)**.** *Let $A, E \in \mathbb{R}^{n \times n}$ be symmetric. Let $\lambda_1, \ldots, \lambda_n$ be the eigenvalues of $A$, arranged in decreasing order. Let $\hat{\lambda}_1, \ldots, \hat{\lambda}_n$ be the eigenvalues of $A + E$, arranged in decreasing order. Then*

$$\sum_{i=1}^{n} \left|\hat{\lambda}_i - \lambda_i\right|^2 \leq \|E\|_F^2 .$$

*Proof.* See, e.g., Horn & Johnson (2012), Chapter 6.3, Theorem 6.3.5 and Corollary 6.3.8. □

**Lemma 1.** *Let $d \geq 4\log(2n^2/\delta)$ for some $\delta \in (0,1)$. Then with probability at least $1 - \delta$, we have*

$$|\langle \bar{x}_i, \bar{x}_j \rangle| < \iota := \widetilde{\mathcal{O}}\left(\frac{1}{\sqrt{d}}\right), \quad i \neq j.$$

*Proof.* See Section C.1. □

By Lemma 1 we can assume $d \geq \text{poly}(n)$ such that $n\iota$ is sufficiently small depends on requirements.

The following two lemmas characterize the projected components of each training data onto the column space of $P_1$, $P_{-1}$, and $P_\perp$.

**Lemma 2.** *For $x_j \neq x_1$, we have*

- $P_{-1}x_j = x_j$;

- $P_1 x_j = 0$;

- $P_\perp x_j = 0$.

*Proof.* These are by the construction of the projection operators. □

**Lemma 3.** *Assume $\sqrt{n}\iota \leq 1/4$. With probability at least $1 - \delta$, we have*

- $0 \leq \|P_{-1}\bar{x}_1\|_2 \leq 2\sqrt{n}\iota$;

- $\sqrt{1 - 4n\iota^2} \leq \|P_1\bar{x}_1\|_2 \leq 1$;

- $P_\perp x_1 = 0$.

*Proof.* See Section C.2. □

The following four lemmas characterize the spectrum of the matrices $H$, $H_{-1}$, $H_1$ and $H_c$.

**Lemma 4.** *Let $\gamma_1, \ldots, \gamma_n$ be the $n$ non-zero eigenvalues of $H := XX^\top$ in decreasing order. then*

$$\lambda_n - n\iota \leq \gamma_1, \ldots, \gamma_n \leq \lambda_1 + n\iota.$$

*Furthermore, if there exist $\lambda_r$ and $\lambda_{r+1}$ such that $\lambda_r > \lambda_{r+1} + 2n\iota$, then*

$$\lambda_n - n\iota \leq \gamma_{r+1}, \ldots, \gamma_n \leq \lambda_{r+1} + n\iota < \lambda_r - n\iota \leq \gamma_1, \ldots, \gamma_r \leq \lambda_1 + n\iota.$$

*Proof.* See Section C.3. □

**Lemma 5.** *Assume $\lambda_n \geq 3n\iota$. Consider the symmetric matrix $H_{-1} := P_{-1}X(P_{-1}X)^\top \in \mathbb{R}^{d \times d}$.*

- *0 is an eigenvalue of $H_{-1}$ with algebraic multiplicity being $d - n + 1$, and its corresponding eigenspace is the column space of $P_1 + P_\perp$.*

- *Restricted in the column space of $P_{-1}$, the $n - 1$ eigenvalues of $H_{-1}$ belong to*

$$(\lambda_n - n\iota, \ \lambda_2 + n\iota).$$

*Proof.* See Section C.4. □

**Lemma 6.** *Consider matrix $H_1 := P_1X(P_1X)^\top \in \mathbb{R}^{d \times d}$. We have $H_1$ has only one non-zero eigenvalue, which belongs to*

$$[\lambda_1 \left(1 - 4n\iota^2\right), \ \lambda_1].$$

*Moreover, the corresponding eigenspace is the column space of $P_1$, which is 1-dim.*

*Proof.* Clearly $H_1$ is rank-1 since the column space of $P_1$ is 1-dim. Thus it has only one non-zero eigenvalue, which is given by

$$\text{tr}(H_1) = \sum_{i=1}^n \|P_1 x_i\|_2^2 = \|P_1 x_1\|_2^2 \in [\lambda_1 \left(1 - 4n\iota^2\right), \ \lambda_1],$$

where the last equality follows from Lemma 3. □

**Lemma 7.** *Consider matrix $H_c := (P_{-1}x_1)(P_1x_1)^\top + (P_1x_1)(P_{-1}x_1)^\top \in \mathbb{R}^{d \times d}$.*

$$\|H_c\|_2 \leq 2\lambda_1 \|P_{-1}\bar{x}_1\|_2 \leq 4\sqrt{n}\iota.$$

*Proof.*
$$\|H_c\|_2 \leq 2 \|P_{-1}x_1\|_2 \cdot \|P_1x_1\|_2 \leq 2\lambda_1 \|P_{-1}\bar{x}_1\|_2 \leq 4\sqrt{n}\iota,$$
where the last equality follows from Lemma 3 and $\lambda_1 \leq 1$.

□

## B  MISSING PROOFS FOR THE THEOREMS IN MAIN TEXT

### B.1  THE DIRECTIONAL BIAS OF SGD WITH MODERATE LEARNING RATE

**Reloading notations**    Let $\pi^k := \left\{\mathcal{B}_1^k, \ldots, \mathcal{B}_m^k\right\}$ be a randomly chosen uniform $m$-partition of $[n]$, where $n = mb$. Then the SGD iterates at the $k$-th epoch can be formulated as:

$$w_{k,j+1} = w_{k,j} - \frac{2\eta_k}{b} \sum_{i \in \mathcal{B}_j^k} x_i x_i^\top (w_{k,j} - w_*), \quad j = 1, \ldots, m,$$

where we assume that the learning rate is fixed within each epoch. Note here $\pi^k$ is independently and randomly chosen at each epoch. For simplicity we often ignore the epoch-indicator $k$, and write the uniform partition as $\pi := \left\{\mathcal{B}_1, \ldots, \mathcal{B}_m\right\}$. It is clear from context that $\pi$ is random over epochs. For a mini-batch $\mathcal{B}_j \in \pi$, we denote $H(\mathcal{B}_j) := \sum_{i \in \mathcal{B}_j} x_i x_i^\top$.

Considering translating the variable by

$$v = w - w_*,$$

then we can reformulate the SGD update rule as

$$v_{k,j+1} = v_{k,j} - \frac{2\eta_k}{b} H(\mathcal{B}_j)v_{k,j} = \left(I - \frac{2\eta_k}{b} H(\mathcal{B}_j)\right) v_{k,j}, \quad j = 1, \ldots, m. \tag{8}$$

Let

$$
\begin{aligned}
\mathcal{M}_\pi &:= \prod_{j=1}^m \left( I - \frac{2\eta_k}{b} H(\mathcal{B}_j) \right) \\
&:= \left( I - \frac{2\eta_k}{b} H(\mathcal{B}_m) \right) \cdot \left( I - \frac{2\eta_k}{b} H(\mathcal{B}_{m-1}) \right) \cdots \left( I - \frac{2\eta_k}{b} H(\mathcal{B}_1) \right).
\end{aligned}
$$

Here the matrix production over a sequence of matrices $\left\{ M_i \in \mathbb{R}^{d \times d} \right\}_{j=1}^m$ is defined from the left to right with descending index,

$$
\prod_{j=1}^m M_j := M_m \times M_{m-1} \times \cdots \times M_1.
$$

Let $v_{k+1} = v_{k,m+1}$ and $v_k = v_{k,1}$. Then we can further reformulate Eq. (8) and obtain the epoch-wise update of SGD as

$$
v_{k+1} = \left( I - \frac{2\eta_k}{b} H(\mathcal{B}_m) \right) \cdot \left( I - \frac{2\eta_k}{b} H(\mathcal{B}_{m-1}) \right) \cdots \left( I - \frac{2\eta_k}{b} H(\mathcal{B}_1) \right) \cdot v_k = \mathcal{M}_\pi v_k. \quad (9)
$$

In light of the notion of $v$, the following lemma restates the related notations of loss functions, hypothesis class, level set, and estimation error defined in Section 2 and 4.

**Lemma 8** (Reloading SGD notations). *Regarding repramaterization $v = w - w_*$, we can reload the following related notations:*

- *Empirical loss and population loss are*

$$
L_\mathcal{S}(v) = \frac{1}{n}(P_1 v)^\top H_1 (P_1 v) + \frac{1}{n}(P_{-1} v)^\top H_{-1}(P_{-1} v) + \frac{1}{n}(Pv)^\top H_c(Pv),
$$
$$
L_\mathcal{D}(v) = \mu \|v\|_2^2.
$$

- *The hypothesis class is*

$$
\mathcal{H}_\mathcal{S} = \left\{ v \in \mathbb{R}^d : P_\perp v = P_\perp v_0 \right\}.
$$

- *The $\alpha$-level set is*

$$
\mathcal{V} = \left\{ v \in \mathcal{H}_\mathcal{S} : L_\mathcal{S}(v) = \alpha \right\}.
$$

- *For $v \in \mathcal{H}_\mathcal{S}$, the estimation error is*

$$
\Delta(v) = \mu \|Pv\|_2^2.
$$

  *Moreover,*

$$
\Delta_* = \inf_{v \in \mathcal{V}} \Delta(v) = \frac{\mu n \alpha}{\gamma_1}.
$$

*Proof.* See Section C.5. $\qquad \square$

Based on the above definitions, the following lemma characterizes the one-step update of SGD.

**Lemma 9** (One step SGD update). *Consider the $j$-th SGD update at the $k$-th epoch as given by Eq. (8). Set the learning rate be constant $\eta$ during that epoch. Then for $j = 1, \dots, m$ we have*

$$
\begin{pmatrix} P_1 v_{k,j+1} \\ P_{-1} v_{k,j+1} \end{pmatrix} = \begin{pmatrix} I - \frac{2\eta}{b} P_1 H(\mathcal{B}_j) P_1 & -\frac{2\eta}{b} P_1 H(\mathcal{B}_j) P_{-1} \\ -\frac{2\eta}{b} P_{-1} H(\mathcal{B}_j) P_1 & I - \frac{2\eta}{b} P_{-1} H(\mathcal{B}_j) P_{-1} \end{pmatrix} \cdot \begin{pmatrix} P_1 v_{k,j} \\ P_{-1} v_{k,j} \end{pmatrix}
$$

*Moreover, if $1 \notin \mathcal{B}_j$, i.e., $x_1$ is not used in the $j$-the step, then*

$$
\begin{pmatrix} P_1 v_{k,j+1} \\ P_{-1} v_{k,j+1} \end{pmatrix} = \begin{pmatrix} I & 0 \\ 0 & I - \frac{2\eta}{b} P_{-1} H(\mathcal{B}_j) P_{-1} \end{pmatrix} \cdot \begin{pmatrix} P_1 v_{k,j} \\ P_{-1} v_{k,j} \end{pmatrix}
$$

*Proof.* See Section C.6. $\qquad \square$

Eq. (9) indicates the key to analyze the convergence of $v_{k+1}$ is to characterize the spectrum of the matrix $\mathcal{M}_\pi$. In particular the following lemma bounds the spectrum of $\mathcal{M}_\pi$ when projected onto the column space of $P_{-1}$.

**Lemma 10.** *Suppose $3n\iota < \lambda_n$. Suppose $0 < \eta < \frac{b}{\lambda_2+3n\iota}$. Let $\pi := \{\mathcal{B}_1, \ldots, \mathcal{B}_m\}$ be a uniform $m$ partition of index set $[n]$, where $n = mb$. Consider the following $d \times d$ matrix*

$$\mathcal{M}_{-1} := \prod_{j=1}^{m} \left( I - \frac{2\eta}{b} P_{-1} H(\mathcal{B}_j) P_{-1} \right) \in \mathbb{R}^{d \times d}.$$

*Then for the spectrum of $\mathcal{M}_{-1}^\top \mathcal{M}_{-1}$ we have:*

- *1 is an eigenvalue of $\mathcal{M}_{-1}^\top \mathcal{M}_{-1}$ with multiplicity being $d - n + 1$; moreover, the corresponding eigenspace is the column space of $P_1 + P_\perp$.*

- *Restricted in the column space of $P_{-1}$, the eigenvalues of $\mathcal{M}_{-1}^\top \mathcal{M}_{-1}$ are upper bounded by $\left(q_{-1}(\eta)\right)^2 < 1$, where*

$$q_{-1}(\eta) := \max \left\{ \left| 1 - \frac{2\eta}{b}(\lambda_2 + n\iota) \right| + \frac{3n\eta\iota}{b}, \ \left| 1 - \frac{2\eta}{b}(\lambda_n - n\iota) \right| + \frac{3n\eta\iota}{b} \right\} < 1.$$

*Proof.* See Section C.7. ☐

Consider the projections of $v_k$ onto the column space of $P_{-1}$ and $P_1$. For simplicity let

$$A_k := \|P_{-1} v_k\|_2, \quad B_k := \|P_1 v_k\|_2.$$

The following lemma controls the update of $A_k$ and $B_k$.

**Lemma 11** (Update rules for $A_k$ and $B_k$). *Suppose $3n\iota < \lambda_n$. Suppose $0 < \eta < \frac{b}{\lambda_2+3n\iota}$. Consider the $k$-th epoch of SGD iterates given by Eq. (9). Set the learning rate in this epoch to be constant $\eta$. Denote*

$$\xi(\eta) := \frac{4\eta\sqrt{n\iota}}{b},$$

$$q_1(\eta) := \left| 1 - \frac{2\eta\lambda_1}{b} \|P_1 \bar{x}_1\|_2^2 \right|,$$

$$q_{-1}(\eta) := \max \left\{ \left| 1 - \frac{2\eta}{b}(\lambda_2 + n\iota) \right| + \frac{3n\eta\iota}{b}, \ \left| 1 - \frac{2\eta}{b}(\lambda_n - n\iota) \right| + \frac{3n\eta\iota}{b} \right\} < 1.$$

*Then we have the following:*

- $A_{k+1} \leq q_{-1}(\eta) \cdot A_k + \xi(\eta) \cdot B_k.$

- $B_{k+1} \leq q_1(\eta) \cdot B_k + \xi(\eta) \cdot A_k.$

- $B_{k+1} \geq q_1(\eta) \cdot B_k - \xi(\eta) \cdot A_k.$

*Proof.* See Section C.8. ☐

Note we can rephrase the update rules for $A_k$ and $B_k$ as

$$\begin{pmatrix} A_{k+1} \\ B_{k+1} \end{pmatrix} \leq \begin{pmatrix} q_{-1}(\eta) & \xi(\eta) \\ \xi(\eta) & q_1(\eta) \end{pmatrix} \cdot \begin{pmatrix} A_k \\ B_k \end{pmatrix},$$

where "$\leq$" means "entry-wisely smaller than".

The following two lemmas characterize the long run behaviors of $A_k$ and $B_k$ with different learning rate.

**Lemma 12** (The long run behavior of SGD with moderate LR). *Suppose $3n\iota < \lambda_n$, and $\lambda_2 + 4n\iota < \lambda_1$. Suppose $v_0$ is far away from $0$. Consider the first $k_1$ epochs of SGD iterates given by Eq. (9). Set the learning rate during this stage to be constant, i.e., $\eta_k = \eta$ for $0 \le k < k_1$. Suppose*

$$\frac{b}{\lambda_1 - 3\sqrt{n\iota}} < \eta < \frac{b}{\lambda_2 + 3n\iota}.$$

*Then for $0 < \epsilon < 1$ and $0 < \beta < \beta_0 < B_0$ satisfying $\sqrt{n\iota} \le \text{poly}(\epsilon\beta)$, there exists $k_1 \ge \mathcal{O}\left(\log \frac{1}{\epsilon\beta}\right)$ such that*

- $A_{k_1} \le \epsilon \cdot \beta.$

- $B_{k_1} \le \|Pv_0\|_2 \cdot \rho_1^{k_1} + \frac{\epsilon}{2} \cdot \beta = \text{poly}\left(\frac{1}{\epsilon\beta}\right).$

- *For all $k = 0, 1, \ldots, k_1$, $B_k > \beta_0$.*

*Proof.* See Section C.9. $\qquad\square$

**Lemma 13** (The long run behavior of SGD with small LR). *Suppose $3n\iota < \lambda_n$, and $\lambda_2 + 4n\iota < \lambda_1$. Suppose $v_0$ is far away from $0$. Consider another $k_2 - k_1$ epochs of SGD iterates given by Eq. (9). Set the learning rate to be constant during the updates, i.e., $\eta_k = \eta'$ for $k_1 \le k < k_2$. Suppose*

$$0 < \eta' < \frac{b}{2\lambda_1}.$$

*Consider the $\epsilon$ and $\beta$ given in Lemma 12. Then for $k \ge k_1$, we have*

- $A_k \le \epsilon \cdot \beta.$

- $B_k \le \begin{cases} q \cdot B_{k-1}, & B_{k-1} > \beta, \\ \beta, & B_{k-1} < \beta, \end{cases}$ *where $q \in (0, 1)$ is a constant.*

*Proof.* See Section C.10. $\qquad\square$

**Theorem 5** (Theorem 1, formal version). *Suppose $3n\iota < \lambda_n$ and $\lambda_2 + 4n\iota < \lambda_1$. Suppose $v_0$ is away from $0$. Consider the SGD iterates given by Eq. (9) with the following moderate learning rate scheme*

$$\eta_k = \begin{cases} \eta \in \left(\frac{b}{\lambda_1 - 3\sqrt{n\iota}}, \frac{b}{\lambda_2 + 3n\iota}\right), & k = 1, \ldots, k_1; \\ \eta' \in \left(0, \frac{b}{2\lambda_1}\right), & k = k_1 + 1, \ldots, k_2. \end{cases}$$

*Then for $0 < \epsilon < 1$ such that $\sqrt{n\iota} \le \text{poly}(\epsilon)$, there exist $k_1 > \mathcal{O}\left(\log \frac{1}{\epsilon}\right)$ and $k_2$ such that*

$$(1 - \epsilon) \cdot \gamma_1 \le \frac{v_{k_2}^\top H v_{k_2}}{\|Pv_{k_2}\|_2^2} \le \gamma_1.$$

*Proof.* We choose $k_1$ and $k_2$ as in Lemma 12 and Lemma 13 with $\beta$ set as a small constant, then we are guaranteed to have

$$A_{k_2} \le \epsilon \cdot \beta \le \epsilon \cdot B_{k_2},$$

from where we have

$$\frac{\|P_1 v_{k_2}\|_2^2}{\|Pv_{k_2}\|_2^2} = \frac{B_{k_2}^2}{A_{k_2}^2 + B_{k_2}^2} \ge \frac{1}{1 + \epsilon^2} \ge 1 - \epsilon^2.$$

Then we have

$$
\frac{v_{k_2}^\top H v_{k_2}}{\|P v_{k_2}\|_2^2} = \frac{(P_1 v_{k_2})^\top H_1 (P_1 v_{k_2})}{\|P v_{k_2}\|_2^2} + \frac{(P_{-1} v_{k_2})^\top H_{-1}(P_{-1}v_{k_2})}{\|P v_{k_2}\|_2^2} + \frac{(P v_{k_2})^\top H_c(P v_{k_2})}{\|P v_{k_2}\|_2^2}
$$

$$
\geq \lambda_1 \left(1 - 4n\iota^2\right) \cdot \frac{\|P_1 v_{k_2}\|_2^2}{\|P v_{k_2}\|_2^2} + 0 - 4\sqrt{n}\iota
$$

$$
\geq \lambda_1 (1 - 4n\iota^2) \cdot (1 - \epsilon^2) - 4\sqrt{n}\iota
$$

$$
\geq (\gamma_1 - n\iota)(1 - 4n\iota^2) \cdot (1 - \epsilon^2) - 4\sqrt{n}\iota \qquad \text{(since } \gamma_1 \leq \lambda_1 + n\iota \text{ by Lemma 4)}
$$

$$
= \gamma_1(1 - 4n\iota^2)(1 - \epsilon^2) - n\iota(1 - 4n\iota^2)(1 - \epsilon^2) - 4\sqrt{n}\iota
$$

$$
\geq \gamma_1(1 - 0.5\epsilon) - 0.5\gamma_1\epsilon \qquad \text{(since } \sqrt{n}\iota \leq \text{poly}(\epsilon))
$$

$$
= \gamma_1(1 - \epsilon).
$$

$\square$

**Theorem 6** (Theorem 4 first part, formal version). *Suppose $3n\iota < \lambda_n$ and $\lambda_2 + 4n\iota < \lambda_1$. Suppose $v_0$ is away from 0. Consider the SGD iterates given by Eq. (9) with the following moderate learning rate schedule*

$$
\eta_k = \begin{cases} \eta \in \left( \frac{b}{\lambda_1 - 3\sqrt{n}\iota}, \ \frac{b}{\lambda_2 + 3n\iota} \right), & k = 1, \ldots, k_1; \\ \eta' \in \left(0, \frac{b}{2\lambda_1}\right), & k = k_1 + 1, \ldots, k_2. \end{cases}
$$

*Then for $0 < \epsilon < 1$ satisfying $\sqrt{n}\iota \leq \text{poly}(\epsilon)$, there exist $k_1$ and $k_2$ such that SGD outputs an $\epsilon$-optimal solution.*

*Proof.* We set

$$
\beta = \sqrt{\frac{n\alpha}{\gamma_1}}, \tag{10}
$$

$$
\beta_0 = \sqrt{\frac{n\alpha}{\gamma_n}} > \beta, \tag{11}
$$

and apply Lemma 12 to choose a $k_1$ such that

$$
\|P_{-1} v_{k_1}\|_2 \leq \epsilon \cdot \beta = \epsilon \cdot \sqrt{\frac{n\alpha}{\gamma_1}}; \tag{12}
$$

$$
\|P_1 v_k\|_2 \geq \beta_0 = \sqrt{\frac{n\alpha}{\gamma_n}}, \quad \forall \ 0 \leq k \leq k_1. \tag{13}
$$

Thus for all $0 \leq k \leq k_1$,

$$
L_{\mathcal{S}}(v_k) = \frac{1}{n}(P v_k)^\top X X^\top (P v_k)
$$

$$
\geq \frac{\gamma_n}{n}\|P v_k\|_2^2 \qquad (\gamma_n \text{ is the smallest eigenvalue of } X X^\top \text{ in the column space of } P)
$$

$$
\geq \frac{\gamma_n}{n}\|P_1 v_k\|_2^2
$$

$$
> \alpha, \qquad \text{(by Eq. (13))}
$$

which implies SGD cannot reach the $\alpha$-level set during the iteration of first stage, i.e., SGD does not terminate in this stage.

We thus consider the second stage. From Lemma 13 we know $\|P_1 v_{k_n}\|_2$ will keep decreasing before being smaller than $\beta$, and $\|P_{-1} v_k\|_2$ stays small during this period, i.e., SGD fits $P_1 v$ while in the same time does not mess up $P_{-1} v$. Mathematically speaking, there exists $k_2$ and $\alpha$ such that

$$
A_{k_2} := \|P_{-1} v_{k_2}\|_2 \leq \epsilon \cdot \beta = \epsilon \cdot \sqrt{\frac{n\alpha}{\gamma_1}},
$$

$$
L_{\mathcal{S}}(v_{k_2}) = \alpha,
$$

which implies SGD terminates at the $k_2$-th epoch. Then by Lemmas 3 and 8, we have

$$
\begin{aligned}
n\alpha &= nL_{\mathcal{S}}(v_{k_2}) \\
&= (P_1 v_{k_2})^\top H_1 (P_1 v_{k_2}) + (P_{-1} v_{k_2})^\top H_2 (P_{-1} v_{k_2}) + (P v_{k_2})^\top H_c (P v_{k_2}) \\
&\geq (P_1 v_{k_2 n})^\top H_1 (P_1 v_{k_2 n}) - \|P_{-1}\bar{x}_1\|_2^2 \cdot \|P v_{k_2 n}\|_2^2 \\
&\geq (\lambda_1 - n\iota) B_{k_2}^2 - 4n\iota^2 (A_{k_2}^2 + B_{k_2}^2) \\
&\geq (\gamma_1 - 3n\iota) B_{k_2}^2 - 4n\iota^2 A_{k_2}^2,
\end{aligned}
$$

which yields

$$
B_{k_2}^2 \leq \frac{n\alpha + 4n\iota^2 A_{k_2}^2}{\gamma_1 - 3n\iota} \leq \left(1 + \frac{\epsilon}{2}\right) \cdot \frac{n\alpha}{\gamma_1}.
$$

Then we can bound the estimation error as

$$
\begin{aligned}
\Delta(v_{k_2}) &= \mu \|P v_{k_2}\|_2^2 \\
&= \mu (B_{k_2}^2 + A_{k_2}^2) \\
&\leq \left(1 + \frac{\epsilon}{2}\right) \cdot \frac{\mu n\alpha}{\gamma_1} + \epsilon^2 \cdot \frac{\mu n\alpha}{\gamma_1} \\
&\leq (1 + \epsilon) \cdot \frac{\mu n\alpha}{\gamma_1} \\
&= (1 + \epsilon) \cdot \Delta_*,
\end{aligned}
$$

where we use the fact that $\Delta_* = \mu n\alpha / \gamma_1$ from Lemma 8. Hence SGD is $\epsilon$-near optimal.

$\square$

## B.2 THE DIRECTIONAL BIAS OF GD WITH MODERATE OR SMALL LEARNING RATE

**Reloading notations**  Denote the eigenvalue decomposition of $XX^\top$ as

$$
XX^\top = G\Gamma G^\top, \quad \Gamma := \mathrm{diag}\,(\gamma_1, \ldots, \gamma_n, 0, \ldots, 0), \quad G = (g_1, \ldots, g_n, \ldots, g_d),
$$

where $G \in \mathbb{R}^{d \times d}$ is orthonormal, and $\gamma_1, \ldots, \gamma_n$ are given by Lemma 4.

Clearly, $\mathrm{span}\,\{g_1, \ldots, g_n\} = \mathrm{span}\,\{x_1, \ldots, x_n\}$. Let

$$
G_\| = (g_1, \ldots, g_n), \quad G_\perp = (g_{n+1}, \ldots, g_d),
$$

then

$$
P = G_\| G_\|^\top, \quad P_\perp = G_\perp G_\perp^\top.
$$

Recall the GD iterates at the $k$-th epoch:

$$
w_{k+1} = w_k - \frac{2\eta_k}{n} XX^\top (w_k - w_*).
$$

Considering translating then rotating the variable as,

$$
u = G^\top (w - w_*),
$$

then we can reformulate the GD iterates as

$$
u_{k+1} = u_k - \frac{2\eta_k}{n} \Gamma u_k = \left(I - \frac{2\eta_k}{n}\Gamma\right) u_k. \tag{14}
$$

We present the following lemma to reload the related notations regarding the parameterization $u = G^\top (w - w_*)$.

**Lemma 14** (Reloading GD notations). *Regarding reparametrization $u = G^\top (w - w_*)$, we can reload the following related notations:*

- *Empirical loss and population loss are*

$$
L_{\mathcal{S}}(u) = \frac{1}{n} \sum_{i=1}^n \gamma_i \left(u^{(i)}\right)^2, \quad L_{\mathcal{D}}(u) = \mu \|u\|_2^2.
$$

- *The hypothesis class is*

$$\mathcal{H}_{\mathcal{S}} = \big\{ u \in \mathbb{R}^d : u^{(i)} = u_0^{(i)}, \text{ for } i = n+1, \dots, d \big\}.$$

- *The $\alpha$-level set is*

$$\mathcal{U} = \big\{ u \in \mathcal{H}_u : L_{\mathcal{S}}(u) = \alpha \big\}.$$

- *For $u \in \mathcal{H}_{\mathcal{S}}$, the estimation error is*

$$\Delta(u) = \mu \sum_{i=1}^{n} \left( u^{(i)} \right)^2.$$

*Moreover,*

$$\Delta_* = \frac{\mu n \alpha}{\gamma_1}.$$

*Proof.* See Section C.11. □

The following lemma sovles GD iterates in Eq. (14).

**Lemma 15.** *For $t = 0, \dots, T$,*

$$u_k^{(i)} = \begin{cases} \prod_{t=0}^{k-1} \left( 1 - \frac{2\eta_t \gamma_i}{n} \right) \cdot u_0^{(i)}, & 1 \le i \le n; \\ u_0^{(i)}, & n+1 \le i \le d. \end{cases}$$

*Proof.* This is by directly solving Eq. (14) where $\Gamma$ is diagonal. □

**Theorem 7** (Theorem 2, formal version)**.** *Suppose $\lambda_n + 2n\iota < \lambda_{n-1}$. Suppose $u_0$ is away from $0$. Consider the GD iterates given by Eq. (14) with learning rate scheme*

$$\eta_k \in \left( 0, \frac{n}{2\lambda_1 + 2n\iota} \right).$$

*Then for $\epsilon \in (0, 1)$, if $k \ge \mathcal{O}\left( \log \frac{1}{\epsilon} \right)$, then we have*

$$\gamma_n \le \frac{u_k^\top \Gamma u_k}{\sum_{i=1}^{n} \left( u_k^{(i)} \right)^2} \le (1 + \epsilon) \cdot \gamma_n.$$

*Proof.* For $i = 1, \dots, n$, denote $q_i(\eta) = 1 - \frac{2\gamma_i}{n} \cdot \eta$, where $\eta \in \left( 0, \frac{n}{2\lambda_1 + 2n\iota} \right)$. Then we have $0 < q_i(\eta) < 1$ since

$$\eta < \frac{n}{2(\lambda_1 + n\iota)} < \frac{n}{2\gamma_1} \le \frac{n}{2\gamma_i},$$

where the second inequality follows from $\gamma_1 < \lambda_1 + n\iota$ by Lemma 4. Furthermore, since $\lambda_n + n\iota < \lambda_{n-1} - n\iota$, Lemma 4 gives us

$$0 < \gamma_n < \gamma_{n-1} \le \dots \le \gamma_1 < 1, \tag{15}$$

which implies

$$1 > q_n(\eta) > q_{n-1}(\eta) \ge \dots \ge q_1(\eta) > 0. \tag{16}$$

Moreover,

$$f(\eta) := \frac{q_{n-1}(\eta)}{q_n(\eta)} = \frac{1 - \frac{2\gamma_{n-1}}{n}\eta}{1 - \frac{2\gamma_n}{n}\eta}$$

is increasing, let $q = \max_{\eta < \frac{n}{2\lambda_1 + 2n\iota}} f(\eta)$, then $q < 1$ by our assumption on the learning rate.

From Lemma 15 we have

$$u_k^{(i)} = \prod_{t=0}^{k-1} q_i(\eta_t) \cdot u_0^{(i)}, \quad i = 1, \dots, n. \tag{17}$$

By the assumption that

$$k > \frac{1}{2} \cdot \frac{\log \frac{\gamma_n \epsilon (u_0^{(n)})^2}{\gamma_1 n \sum_{i=1}^n \left(u_0^{(i)}\right)^2}}{\log q} = \mathcal{O}\left(\frac{1}{\epsilon}\right), \tag{18}$$

we have

$$\frac{\sum_{i=1}^n \left(u_k^{(i)}\right)^2}{(u_k^{(n)})^2} = 1 + \sum_{i=1}^{n-1} \frac{(u_k^{(i)})^2}{(u_k^{(n)})^2}$$

$$= 1 + \sum_{i=1}^{n-1} \frac{\prod_{t=0}^{k-1} q_i(\eta_t)^2 \cdot (u_0^{(i)})^2}{\prod_{t=0}^{k-1} q_n(\eta_t)^2 \cdot (u_0^{(n)})^2} \qquad \text{(by Eq. (17))}$$

$$\leq 1 + \frac{\sum_{i=1}^n \left(u_0^{(i)}\right)^2}{(u_0^{(n)})^2} \cdot \sum_{i=1}^{n-1} \prod_{t=0}^{k-1} \frac{q_i(\eta_t)^2}{q_n(\eta_t)^2}$$

$$\leq 1 + \frac{\sum_{i=1}^n \left(u_0^{(i)}\right)^2}{(u_0^{(n)})^2} \cdot n \cdot \prod_{t=0}^{k-1} \frac{q_{n-1}(\eta_t)^2}{q_n(\eta_t)^2} \qquad \text{(by Eq. (16))}$$

$$\leq 1 + \frac{\sum_{i=1}^n \left(u_0^{(i)}\right)^2}{(u_0^{(n)})^2} \cdot n \cdot q^{2k}$$

$$\leq 1 + \frac{\gamma_n}{\gamma_1} \epsilon, \qquad \text{(by Eq. (18))}$$

which further yields

$$1 \geq \frac{(u_k^{(n)})^2}{\sum_{i=1}^n \left(u_k^{(i)}\right)^2} \geq \frac{1}{1 + \frac{\gamma_n}{\gamma_1}\epsilon} \geq 1 - \frac{\gamma_n}{\gamma_1}\epsilon. \tag{19}$$

By the above inequalities we have

$$\frac{u_k^\top \Gamma u_k}{\sum_{i=1}^n \left(u_k^{(i)}\right)^2} = \sum_{i=1}^n \frac{(u_k^{(i)})^2}{\sum_{i=1}^n \left(u_k^{(i)}\right)^2} \cdot \gamma_i$$

$$= \frac{(u_k^{(n)})^2}{\sum_{i=1}^n \left(u_k^{(i)}\right)^2} \cdot \gamma_n + \sum_{i=1}^{n-1} \frac{(u_k^{(i)})^2}{\sum_{i=1}^n \left(u_k^{(i)}\right)^2} \cdot \gamma_i$$

$$\leq \gamma_n + \sum_{i=1}^{n-1} \frac{(u_k^{(i)})^2}{\sum_{i=1}^n \left(u_k^{(i)}\right)^2} \cdot \gamma_1 \qquad \text{(by Eq. (15))}$$

$$= \gamma_n + \left(1 - \frac{(u_k^{(n)})^2}{\sum_{i=1}^n \left(u_k^{(i)}\right)^2}\right) \cdot \gamma_1$$

$$\leq \gamma_n + \frac{\gamma_n}{\gamma_1}\epsilon \cdot \gamma_1 \qquad \text{(by Eq. (19))}$$

$$= \gamma_n \cdot (1 + \epsilon).$$

Finally we note that $\frac{u_k^\top \Gamma u_k}{\sum_{i=1}^n \left(u_k^{(i)}\right)^2} \geq \gamma_n$ since $\gamma_n$ is the smallest in $\{\gamma_i\}_{i=1}^n$. □

**Theorem 8** (Theorem 4 second part, formal version). *Suppose $\lambda_n + 2n\iota < \lambda_{n-1}$. Suppose $u_0$ is away from $0$. Consider the GD iterates given by Eq. (14) with learning rate scheme*

$$\eta_k \in \left(0, \frac{n}{2\lambda_1 + 2n\iota}\right).$$

*Then for $\epsilon \in (0,1)$, if $k \geq \mathcal{O}\left(\log \frac{1}{\epsilon}\right)$ ,, then GD outputs an $M$-suboptimal solution, where $M = \frac{\gamma_1}{\gamma_n}(1-\epsilon) > 1$ is a constant.*

*Proof.* Consider an $\alpha$-level set where

$$\alpha = L_{\mathcal{S}}(u_k) = \frac{1}{n} u_k^\top \Gamma u_k. \tag{20}$$

From Lemma 15 we know $L_{\mathcal{S}}(u_k)$ is monotonic decreasing thus GD cannot terminate before the $k$-epoch, i.e., the output of GD is $u_k$.

Thus

$$
\begin{aligned}
\frac{\Delta(u)}{\Delta_*} &= \gamma_1 \frac{\sum_{i=1}^n \left(u_k^{(i)}\right)^2}{n\alpha} && \text{(by Lemma 14)} \\
&= \gamma_1 \cdot \frac{\sum_{i=1}^n \left(u_k^{(i)}\right)^2}{u_k^\top \Gamma u_k} && \text{(by Eq. (20))} \\
&\geq \gamma_1 \cdot \frac{1}{(1+\epsilon)\gamma_n} && \text{(by Theorem 7)} \\
&\geq \frac{\gamma_1}{\gamma_n}(1-\epsilon) \\
&=: M,
\end{aligned}
$$

where we have $M > 1$ by letting $\epsilon < 1 - \frac{\gamma_n}{\gamma_1}$. $\qquad\square$

### B.3 THE DIRECTIONAL BIAS OF SGD WITH SMALL LEARNING RATE

We analyze SGD with small learning rate by repeating the arguments in previous two sections.

Let us denote $X_{-n} := (x_1, x_2, \ldots, x_{n-1})$ and

$$P_{-n} = X_{-n}(X_{-n}^\top X_{-n})^{-1} X_{-n}^\top$$
$$P_n = P - P_{-n}.$$

That is, $P_{-n}$ is the projection onto the column space of $X_{-n}$ and $P_n$ is the projection onto the orthogonal complement of the column space of $X_{-n}$ with respect to the column space of $X$.

Let us reload

$$
\begin{aligned}
H &:= XX^\top, \\
H_{-n} &:= (P_{-n}X)(P_{-n}X)^\top, \\
H_n &:= (P_n X)(P_n X)^\top, \\
H_c &:= (P_{-n}x_n)(P_n x_n)^\top + (P_n x_n)(P_{-n}x_n)^\top.
\end{aligned}
$$

Then

$$H = H_{-n} + H_n + H_c.$$

Following a routine check we can reload the following lemmas.

**Lemma 16** (Variant of Lemma 10). *Suppose $3n\iota < \lambda_n$. Suppose $0 < \eta < \frac{b}{\lambda_1 + 3n\iota}$. Let $\pi := \{\mathcal{B}_1, \ldots, \mathcal{B}_m\}$ be a uniform $m$ partition of index set $[n]$, where $n = mb$. Consider the following $d \times d$ matrix*

$$\mathcal{M}_{-n} := \prod_{j=1}^m \left(I - \frac{2\eta}{b} P_{-n} H(\mathcal{B}_j) P_{-n}\right) \in \mathbb{R}^{d \times d}.$$

*Then for the spectrum of $\mathcal{M}_{-n}^\top \mathcal{M}_{-n}$ we have:*

- *1 is an eigenvalue of $\mathcal{M}_{-n}^\top \mathcal{M}_{-n}$ with multiplicity being $d - n + 1$; moreover, the corresponding eigenspace is the column space of $P_n + P_\perp$.*

- *Restricted in the column space of $P_{-n}$, the eigenvalues of $\mathcal{M}_{-n}^{\top}\mathcal{M}_{-n}$ are upper bounded by $(q_{-n}(\eta))^2 < 1$, where*

$$q_{-n}(\eta) := \max\left\{\left|1 - \frac{2\eta}{b}(\lambda_1 + n\iota)\right| + \frac{3n\eta\iota}{b}, \ \left|1 - \frac{2\eta}{b}(\lambda_{n-1} - n\iota)\right| + \frac{3n\eta\iota}{b}\right\} < 1.$$

*Proof.* This is by a routine check of the proof of Lemma 10. $\qquad\square$

Consider the projections of $v_k$ onto the column space of $P_{-n}$ and $P_n$. For simplicity we reload the following notations

$$A_k := \|P_{-n}v_k\|_2, \quad B_k := \|P_n v_k\|_2.$$

The following lemma controls the update of $A_k$ and $B_k$.

**Lemma 17** (Variant of Lemma 11). *Suppose $3n\iota < \lambda_n$. Suppose $0 < \eta < \frac{b}{\lambda_1 + 3n\iota}$. Consider the $k$-th epoch of SGD iterates given by Eq. (9). Set the learning rate in this epoch to be constant $\eta$. Denote*

$$\xi(\eta) := \frac{4\eta\sqrt{n}\iota}{b},$$

$$q_n(\eta) := \left|1 - \frac{2\eta\lambda_n}{b}\|P_n\bar{x}_n\|_2^2\right|,$$

$$q_{-n}(\eta) := \max\left\{\left|1 - \frac{2\eta}{b}(\lambda_1 + n\iota)\right| + \frac{3n\eta\iota}{b}, \ \left|1 - \frac{2\eta}{b}(\lambda_{n-1} - n\iota)\right| + \frac{3n\eta\iota}{b}\right\} < 1.$$

*Then we have the following:*

- $A_{k+1} \le q_{-n}(\eta) \cdot A_k + \xi(\eta) \cdot B_k.$

- $B_{k+1} \le q_n(\eta) \cdot B_k + \xi(\eta) \cdot A_k.$

- $B_{k+1} \ge q_n(\eta) \cdot B_k - \xi(\eta) \cdot A_k.$

*Proof.* This is by a routine check of the proof of Lemma 11. $\qquad\square$

**Lemma 18** (Variant of Lemma 13). *Suppose $3n\iota < \lambda_n$ and $\lambda_n + 4n\iota < \lambda_{n-1}$. Consider the SGD iterates given by Eq. (9) with the following small learning rate scheme*

$$\eta_k = \eta' \in \left(0, \ \frac{b}{2\lambda_1 + 2n\iota}\right), \quad k = 1, \ldots, k_2.$$

*Then for $0 < \epsilon < 1$ satisfying $\sqrt{n}\iota \le \text{poly}(\epsilon)$, if $k_2 \ge \mathcal{O}\left(\log\frac{1}{\epsilon}\right)$, then $\frac{A_{k_2}}{B_{k_2}} \le \epsilon$.*

*Proof.* From the assumption we have $\eta' < \frac{b}{2(\lambda_1 + n\iota)}$ and $\eta' < \frac{b}{2\lambda_n}$, thus

$$\xi := \xi(\eta') = \frac{4\eta'\sqrt{n}\iota}{b},$$

$$q_n := q_n(\eta') = \left| 1 - \frac{2\eta'\lambda_n}{b}\|P_n\bar{x}_n\|_2^2 \right|$$

$$= 1 - \frac{2\eta'\lambda_n}{b}\|P_n\bar{x}_n\|_2^2$$

$$\leq 1 - \frac{2\lambda_n(1 - 4n\iota^2)}{b}\eta', \qquad \text{(since } \|P_n\bar{x}_n\|_2^2 \geq 1 - 4n\iota^2 \text{ by reloading Lemma 3 )}$$

$$< 1$$

$$q_{-n} := q_{-n}(\eta') = \max\left\{ \left| 1 - \frac{2\eta'}{b}(\lambda_1 + n\iota) \right| + \frac{3n\eta'\iota}{b}, \left| 1 - \frac{2\eta'}{b}(\lambda_{n-1} - n\iota) \right| + \frac{3n\eta'\iota}{b} \right\}$$

$$= \max\left\{ 1 - \frac{2\eta'}{b}(\lambda_1 + n\iota) + \frac{3n\eta'\iota}{b}, \ 1 - \frac{2\eta'}{b}(\lambda_{n-1} - n\iota) + \frac{3n\eta'\iota}{b} \right\}$$

$$= 1 - \frac{2\eta'}{b}(\lambda_{n-1} - n\iota) + \frac{3n\eta'\iota}{b}$$

$$= 1 - \frac{2(\lambda_{n-1} - n\iota) - 3n\iota}{b}\eta' \in (0, 1).$$

Moreover, by the gap assumption $\lambda_n + 4n\iota < \lambda_{n-1}$ we have

$$q_n - q_{-n} \geq \eta'\left( \frac{2(\lambda_{n-1} - n\iota) - 3n\iota}{b} - \frac{2\lambda_n(1 - 4n\iota^2)}{b} \right)$$

$$\geq \frac{2\eta'}{b}(\lambda_{n-1} - \lambda_n - 3n\iota)$$

$$> 0.$$

Therefore $0 < q_{-n} < q_n < 1$. Thus we can set $\xi = \frac{4\eta'\sqrt{n}\iota}{b}$ to be small such that

$$0 < q := \frac{q_{-n}}{q_n - \xi \cdot A_0/B_0} < 1. \tag{21}$$

Moreover, since $\sqrt{n}\iota \leq \text{poly}(\epsilon)$ and $\xi = \frac{4\eta'\sqrt{n}\iota}{b}$, we have

$$\frac{\xi}{q_n - \xi \cdot A_0/B_0} \leq \frac{(1-q)\epsilon}{2}. \tag{22}$$

Now we recursively show $\frac{A_k}{B_k} \leq \frac{A_0}{B_0}$. Clearly it holds for $k = 0$. Suppose $\frac{A_k}{B_k} \leq \frac{A_0}{B_0}$, we consider $\frac{A_{k+1}}{B_{k+1}}$ in the following

$$\frac{A_{k+1}}{B_{k+1}} \leq \frac{q_{-n} \cdot A_k + \xi \cdot B_k}{q_n \cdot B_k - \xi \cdot A_k} \qquad \text{(by Lemma 17)}$$

$$= \frac{q_{-n} \cdot \frac{A_k}{B_k} + \xi}{q_n - \xi \cdot \frac{A_k}{B_k}}$$

$$\leq \frac{q_{-n}\frac{A_k}{B_k} + \xi}{q_n - \xi \cdot A_0/B_0} \qquad \text{(by inductive assumption)}$$

$$= \frac{q_{-n}}{q_n - \xi \cdot A_0/B_0}\frac{A_k}{B_k} + \frac{\xi}{q_n - \xi \cdot A_0/B_0}$$

$$\leq q \cdot \frac{A_k}{B_k} + \frac{(1-q)\epsilon}{2} \qquad \text{(by Eq. (21) and (22))}$$

$$\leq q \cdot \frac{A_0}{B_0} + \frac{(1-q)\epsilon}{2}$$

$$\leq \frac{A_0}{B_0},$$

where in the last inequality we assume $\frac{\epsilon}{2} < \frac{A_0}{B_0}$.

Moreover, from the above we have

$$\frac{A_{k+1}}{B_{k+1}} \leq q \cdot \frac{A_k}{B_k} + \frac{(1-q)\epsilon}{2},$$

which implies

$$\frac{A_{k_2}}{B_{k_2}} \leq q^{k_2} \cdot \frac{A_0}{B_0} + \frac{1}{1-q} \cdot \frac{(1-q)\epsilon}{2},$$
$$\leq \frac{\epsilon}{2} + \frac{\epsilon}{2} = \epsilon,$$

where we set $k_2 \geq \mathcal{O}\left(\log \frac{1}{\epsilon}\right)$.

$\square$

Next we prove the directional bias of SGD with small learning rate.

**Theorem 9** (Theorem 3, formal version). *Suppose $3n\iota < \lambda_n$ and $\lambda_n + 4n\iota < \lambda_{n-1}$. Suppose $v_0$ is away from $0$. Consider the SGD iterates given by Eq. (9) with the following small learning rate scheme*

$$\eta_k = \eta' \in \left(0, \frac{b}{2\lambda_1 + 2n\iota}\right), \quad k = 1, \dots, k_2.$$

*Then for $0 < \epsilon < 1$ satisfying $\sqrt{n}\iota \leq \text{poly}(\epsilon)$, if $k_2 \geq \mathcal{O}\left(\log \frac{1}{\epsilon}\right)$, then*

$$\gamma_n \leq \frac{v_{k_2}^\top H v_{k_2}}{\|Pv_{k_2}\|_2^2} \leq (1+\epsilon) \cdot \gamma_n.$$

*Proof.* First by Lemma 18 we have

$$\frac{B_{k_2}^2}{A_{k_2}^2 + B_{k_2}^2} = \frac{1}{\frac{A_{k_2}^2}{B_{k_2}^2} + 1} \geq \frac{1}{\epsilon^2 + 1} \geq 1 - \epsilon^2. \tag{23}$$

Next by $H = H_n + H_{-n} + H_c$ we obtain

$$\frac{v_{k_2}^\top H v_{k_2}}{\|Pv_{k_2}\|_2^2} = \frac{(P_n v_{k_2})^\top H_n (P_n v_{k_2})}{\|Pv_{k_2}\|_2^2} + \frac{(P_{-n} v_{k_2})^\top H_{-n} (P_{-n} v_{k_2})}{\|Pv_{k_2}\|_2^2} + \frac{(Pv_{k_2})^\top H_c (Pv_{k_2})}{\|Pv_{k_2}\|_2^2}$$
$$\leq \lambda_n \cdot \frac{\|P_n v_{k_2}\|_2^2}{\|Pv_{k_2}\|_2^2} + (\lambda_1 + n\iota) \cdot \frac{\|P_{-n} v_{k_2}\|_2^2}{\|Pv_{k_2}\|_2^2} + 4\sqrt{n}\iota \quad \text{by reloading Lemma 5, 6, 7}$$
$$\leq \lambda_n + (\lambda_1 + n\iota) \cdot \frac{A_{k_2}^2}{A_{k_2}^2 + B_{k_2}^2} + 4\sqrt{n}\iota$$
$$\leq \gamma_n + n\iota + (\lambda_1 + n\iota) \cdot \epsilon^2 + 4\sqrt{n}\iota \quad \text{by reloading Lemma 4 and Eq. (23)}$$
$$\leq \gamma_n + \gamma_n \cdot \epsilon. \quad \text{since } \sqrt{n}\iota \leq \text{poly}(\epsilon)$$

Finally, we note $\frac{v_{k_2}^\top H v_{k_2}}{\|Pv_{k_2}\|_2^2} \geq \gamma_n$ since $\gamma_n$ is the smallest eigenvalue of $H$ restricted in the column space of $P$. $\square$

**Theorem 10** (Theorem 4 third part, formal version). *Suppose $3n\iota < \lambda_n$ and $\lambda_n + 4n\iota < \lambda_{n-1}$. Suppose $v_0$ is away from $0$. Consider the SGD iterates given by Eq. (9) with the following small learning rate scheme*

$$\eta_k = \eta' \in \left(0, \frac{b}{2\lambda_1 + 2n\iota}\right), \quad k = 1, \dots, k_2.$$

*Then for $0 < \epsilon < 1$ such that $\sqrt{n}\iota \leq \text{poly}(\epsilon)$, if $k_2 \geq \mathcal{O}\left(\log \frac{1}{\epsilon}\right)$, then SGD outputs an $M$-suboptimal solution where $M = \frac{\gamma_1}{\gamma_n}(1-\epsilon) > 1$ is a constant.*

*Proof.* From Eq. (9) and $\eta' < \frac{1}{2\lambda_1}$ we know that restricted in the column space of $P$, the eigenvalues of $\mathcal{M}_\pi$ is smaller than 1, thus $v_k$ indeed converges to 0.

Consider an $\alpha$-level set where

$$\alpha = L_{\mathcal{S}}(v_k) = \frac{1}{n} v_{k_2}^\top H v_{k_2}. \tag{24}$$

Then

$$\begin{aligned}
\frac{\Delta(u)}{\Delta_*} &= \gamma_1 \frac{\|Pv_k\|_2^2}{n\alpha} && \text{(by Lemma 8)} \\
&= \gamma_1 \cdot \frac{\|Pv_k\|_2^2}{v_k^\top H v_k} && \text{(by Eq. (24))} \\
&\geq \gamma_1 \cdot \frac{1}{(1+\epsilon)\gamma_n} && \text{(by Theorem 9)} \\
&\geq \frac{\gamma_1}{\gamma_n}(1-\epsilon) \\
&=: M,
\end{aligned}$$

where we have $M > 1$ by letting $\epsilon < 1 - \frac{\gamma_n}{\gamma_1}$. $\qquad\square$

## C  Proof of Auxiliary Lemmas in Sections A and B

### C.1  Proof of Lemma 1

*Proof of Lemma 1.* Note that $\bar{x}_i$ follows uniform distribution on the sphere $\mathcal{S}^{d-1}$. Therefore, let $\xi$ be a random variable following distribution $\chi_d^2$ distribution and define $z_i = \xi \cdot \bar{x}_i$, we have $z_i$ follows standard normal distribution in the $d$-dimensional space. Then it suffices to prove that $|\langle z_i, z_j \rangle|/(\|z_i\|_2\|z_j\|_2) \leq \iota$ for all $i \neq j$.

First we will bound the inner product $\langle z_i, z_j \rangle$. Note that we have each entry in $z_i$ is 1-subgaussian, it can be direcly deduced that

$$\langle z_i, z_j \rangle = \sum_{k=1}^d z_i^{(k)} z_j^{(k)} = \sum_{k=1}^d \left( \left( \frac{z_i^{(k)} + z_j^{(k)}}{2} \right)^2 - \left( \frac{z_i^{(k)} - z_j^{(k)}}{2} \right)^2 \right)$$

is $d$-subexponential, where $z_i^{(k)}$ denotes the $k$-th of the vector $z_i$. Then if follows that

$$\mathbb{P}\left( |\langle z_i, z_j \rangle| \geq t \right) \leq 2 \exp\left( -\frac{t^2}{d} \right).$$

Next we will lower bound $\|z_i\|_2$. Note that

$$\|z_i\|_2^2 - d = \sum_{k=1}^d \left( \left( z_i^{(k)} \right)^2 - 1 \right).$$

Since $z_i^{(k)}$ is 1-subgaussian, we have $\|z_i\|^2 - d$ is $d$-subexpoential, then

$$\mathbb{P}\left( \left| \|z_i\|_2^2 - d \right| \geq t \right) \leq 2 \exp\left( -\frac{t^2}{d} \right).$$

Finally, applying the union bound for all possible $i, j \in [n]$, we have with probability at least $1 - \delta$, the following holds for all $i \neq j$,

$$|\langle z_i, z_j \rangle| \leq \sqrt{d \log \frac{2n^2}{\delta}},$$

$$\|z_i\|_2^2 \geq d - \sqrt{d \log \frac{2n^2}{\delta}}.$$

Assume $d \geq 4 \log(2n^2/\delta)$, we have $\|z_i\|^2 \geq d/2$. Then it follows that

$$|\langle \bar{x}_i, \bar{x}_j \rangle| = \frac{|\langle z_i, z_j \rangle|}{\|z_i\|_2 \|z_j\|_2} < 2\sqrt{\frac{1}{d} \log \frac{2n^2}{\delta}} =: \iota.$$

This completes the proof. $\qquad\square$

## C.2 PROOF OF LEMMA 3

*Proof of Lemma 3.* Similar to the proof of Lemma 1, we consider translating $x_1, \ldots, x_n$ to $z_1, \ldots, z_n$ by introducing $\chi_d^2$ random variables. Let $Z_{-1} = (z_2, \ldots, z_n) \in \mathbb{R}^{d \times (n-1)}$, in which each entry is i.i.d. generated from Gaussian distribution $\mathcal{N}(0,1)$. Then we have

$$P_{-1}\bar{x}_1 = X_{-1}(X_{-1}^\top X_{-1})^{-1}X_{-1}^\top \bar{x}_1 = Z_{-1}(Z_{-1}^\top Z_{-1})^{-1}Z_{-1}^\top \bar{x}_1.$$

Then conditioned on $\bar{x}_1$, we have each entry in $Z_{-1}^\top \bar{x}_1$ i.i.d. follows $\mathcal{N}(0,1)$. Then it is clear that $\|Z_{-1}^\top \bar{x}_1\|_2^2$ follows from $\chi_{n-1}^2$ distribution, implying that with probability at least $1 - \delta'$, we have

$$\|Z_{-1}^\top \bar{x}_1\|_2^2 \leq (n-1) + \sqrt{(n-1)\log(2/\delta')}.$$

Then by Corollary 5.35 in Vershynin (2010), we know that with probability at least $1 - \delta'$ it holds that

$$\sqrt{d} - \sqrt{n-1} - \sqrt{2\log(2/\delta')} \leq \sigma_{\min}(Z_{-1}) \leq \sigma_{\max}(Z_{-1}) \leq \sqrt{d} + \sqrt{n-1} + \sqrt{2\log(2/\delta')}.$$

Therefore, assume $\sqrt{(n-1)} + \sqrt{2\log(2/\delta')} \leq \sqrt{d}/8$, we have with probability at least $1 - \delta'$

$$\left\|Z_{-1}(Z_{-1}^\top Z_{-1})^{-1}\right\|_2 \leq \frac{\sqrt{d} + \sqrt{n-1} + \sqrt{2\log(2/\delta')}}{\left(\sqrt{d} - \sqrt{n-1} - \sqrt{2\log(2/\delta')}\right)^2}$$

$$\leq \frac{1}{\sqrt{d}}\left(1 + 4\left(\sqrt{\frac{n-1}{d}} + \sqrt{\frac{2\log(2/\delta')}{d}}\right)\right).$$

Combining with the upper bound of $\|Z_{-1}^\top \bar{x}_1\|_2$, set $\delta' = \delta/2$, we have with probability at least $1 - \delta$ that

$$\|P_{-1}\bar{x}_1\|_2 \leq \left\|Z_{-1}(Z_{-1}^\top Z_{-1})^{-1}\right\|_2 \cdot \|X_{-1}^\top \bar{x}_1\|_2$$

$$\leq \left(1 + 4\left(\sqrt{\frac{n-1}{d}} + \sqrt{\frac{2\log(4/\delta)}{d}}\right)\right) \cdot \left(\sqrt{\frac{n-1}{d}} + \sqrt{\frac{\sqrt{(n-1)\log(4/\delta)}}{d}}\right)$$

$$\leq (1 + 4\sqrt{n}\iota) \cdot \sqrt{n}\iota,$$

where the last inequality follows from the definition of $\iota$. Then assume $\sqrt{n}\iota \leq 1/4$, we are able to completes the proof of the first argument. Note that $\|P_1\bar{x}_1\|_2^2 + \|P_{-1}\bar{x}_1\|_2^2 = \|\bar{x}_1\|_2^2 = 1$, we have

$$\|P_{-1}\bar{x}_1\|_2 = \sqrt{1 - \|P_1\bar{x}_1\|_2^2} \geq \sqrt{1 - 4n\iota^2} \geq 1 - 4n\iota^2.$$

This completes the proof of the second argument. The third argument holds trivially by the construction of $P_\perp$.

$\square$

## C.3 PROOF OF LEMMA 4

*Proof of Lemma 4.* Clearly $XX^\top \in \mathbb{R}^{d \times d}$ is of rank $n$ and symmetric, thus $XX^\top$ has $n$ real, non-zero (potentially repeated) eigenvalues, denoted as $\gamma_1, \ldots, \gamma_n$ in non-decreasing order. Moreover, $\gamma_1, \ldots, \gamma_n$ are also eigenvalues of $X^\top X \in \mathbb{R}^{n \times n}$, thus it is sufficient to locate the eigenvalues of $X^\top X$, where $(X^\top X)_{ij} = x_i^\top x_j$.

We first calculate the diagonal entry

$$\left(X^\top X\right)_{ii} = x_i^\top x_i = \lambda_i.$$

Then we bound the off diagonal entries. For $j \neq i$,

$$\left(X^\top X\right)_{ij} = x_i^\top x_j = \sqrt{\lambda_i \lambda_j}\langle \bar{x}_i, \bar{x}_j \rangle \in (-\iota, \iota),$$

where we use $0 < \lambda_1, \ldots, \lambda_n \leq 1$. Thus we have

$$R_i(X^\top X) = \sum_{j \neq i}\left|\left(X^\top X\right)_{ij}\right| \leq n\iota, \quad i = 1, \ldots, n,$$

Finally our conclusions hold by applying Gershgorin circle theorem.

$\square$

## C.4    PROOF OF LEMMA 5

*Proof of Lemma 5.* The first conclusion is clear since by construction, we have $P_{-1}P_1 = P_{-1}P_\perp = 0$.

Note that $H_{-1}$ is a rank $n-1$ symmetric matrix. Let $\tau_2, \ldots, \tau_n$ be the $n-1$ non-zero eigenvalues of $H_{-1}$. Clearly, $\tau_2, \ldots, \tau_n$ with $\tau_1 := 0$ give the spectrum of

$$H'_{-1} := (P_{-1}X)^\top P_{-1}X \in \mathbb{R}^{n \times n}.$$

We then bound $\tau_2, \ldots, \tau_n$ by analyzing $H'_{-1}$.

From Lemma 3 we have $\|P_{-1}\bar{x}_1\|_2 \le 2\sqrt{n}\iota$. From Lemma 2 we have

$$P_{-1}X = (P_{-1}x_1, P_{-1}x_2, \ldots, P_{-1}x_n) = (P_{-1}x_1, x_2, \ldots, x_n).$$

Then we calculate the diagonal entries:

$$\left(H'_{-1}\right)_{ii} = \begin{cases} \|P_{-1}x_1\|_2^2 \le \lambda_1 \cdot 4n\iota^2 \le 4n\iota^2, & i = 1; \\ \|x_i\|_2^2 = \lambda_i, & i \neq 1. \end{cases}$$

Then we bound the off diagonal entries. Let $j \neq i$. Then at least one of them is not 1. Without loss of generality let $i \neq 1$, which yields $x_i = P_{-1}x_i$ by Lemma (2). Thus $\langle x_i, P_1 x_j \rangle = \langle P_{-1}x_i, P_1 x_j \rangle = 0$. Thus we have

$$\begin{aligned}
\left(H'_{-1}\right)_{ij} &= (P_{-1}x_i)^\top P_{-1}x_j \\
&= x_i^\top P_{-1}x_j \\
&= x_i^\top x_j - x_i^\top P_1 x_j \\
&= x_i^\top x_j \\
&= \sqrt{\lambda_i \lambda_j} \cdot \langle \bar{x}_i, \bar{x}_j \rangle \\
&\in (-\iota, \iota).
\end{aligned}$$

Thus we have

$$R_i(H'_{-1}) = \sum_{j \neq i} \left|\left(H'_{-1}\right)_{ij}\right| \le n\iota, \quad i = 1, \ldots, n.$$

Finally, we set $4n\iota^2 + 2n\iota < \lambda_n$, so that the first Geoshgorin disc does not intersect with the others, then Gershgorin circle theorem gives our second conclusion. $\square$

## C.5    PROOF OF LEMMA 8

*Proof of Lemma 8.* For the empirical loss, it is clear that

$$\begin{aligned}
L_{\mathcal{S}}(v) &= \frac{1}{n}(w - w_*)^\top XX^\top (w - w_*) = \frac{1}{n}v^\top XX^\top v = \frac{1}{n}v^\top Hv \\
&= \frac{1}{n}(Pv)^\top H(Pv) \\
&= \frac{1}{n}(Pv)^\top H_1(Pv) + \frac{1}{n}(Pv)^\top H_{-1}(Pv) + \frac{1}{n}(Pv)^\top H_c(Pv) \\
&= \frac{1}{n}(P_1 v)^\top H_1(P_1 v) + \frac{1}{n}(P_{-1}v)^\top H_{-1}(P_{-1}v) + \frac{1}{n}(Pv)^\top H_c(Pv),
\end{aligned}$$

where we use Lemma 4, Lemma 5, and Lemma 6. For the population loss,

$$L_{\mathcal{D}}(v) = \mu \|w - w_*\|_2^2 = \mu \|v\|_2^2.$$

For the hypothesis class $\mathcal{H}_{\mathcal{S}} = \{w \in \mathbb{R}^d : P_\perp w = P_\perp w_0\}$, applying $w - w_* = v$ and $w_0 - w_* = v_0$, we obtain

$$\mathcal{H}_{\mathcal{S}} = \{v \in \mathbb{R}^d : P_\perp v = P_\perp v_0\}.$$

For the $\alpha$-level set, we note the optimal training loss is $L_{\mathcal{S}}^* = \inf_{v \in \mathcal{H}_{\mathcal{S}}} L_{\mathcal{S}}(v) = 0$.

As for the estimation error, we note that $\inf_{v \in \mathcal{H}_\mathcal{S}} L_\mathcal{D}(v) = \inf_{P_\perp v = P_\perp v_0} \mu \|v\|_2^2 = \mu \|P_\perp v_0\|_2^2$. thus for $v \in \mathcal{H}_\mathcal{S}$, we have

$$\Delta(v) = L_\mathcal{D}(v) - \inf_{v' \in \mathcal{V}} L_\mathcal{D}(v') = \mu \|v\|_2^2 - \mu \|P_\perp v_0\|_2^2 = \mu \|Pv\|_2^2.$$

Finally, consider $v \in \mathcal{V}$, i.e., $n\alpha = v^\top X X^\top v$, thus

$$\Delta_* = \inf_{v \in \mathcal{V}} \Delta(v) = \inf_{n\alpha = v^\top X X^\top v} \mu \|Pv\|_2^2 = \frac{\mu n \alpha}{\gamma_1},$$

where $\gamma_1$ is the largest eigenvalue of the matrix $XX^\top$ and the inferior is attended by setting $v$ parallel to the first eigenvector of $XX^\top$. $\qquad\square$

## C.6   PROOF OF LEMMA 9

*Proof of Lemma 9.* From Eq. (8) we have

$$v_{k,j+1} = \left(I - \frac{2\eta}{b} H(\mathcal{B}_j)\right) v_{k,j}, \quad j = 1, \dots, m. \tag{25}$$

Recall the following property of projection operators:

$$P_1 = P_1 P_1, \quad P_{-1} = P_{-1} P_{-1}$$
$$0 = P_1 P_{-1} = P_{-1} P_1.$$

Moreover since $x_i^\top P_\perp v = 0$, we have

$$H(\mathcal{B}_j) P_\perp v = \sum_{i \in \mathcal{B}_j} x_i x_i^\top P_\perp v = 0.$$

Applying $P_1$ to Eq. (25) we have

$$P_1 v_{k,j+1} = P_1 \left(I - \frac{2\eta}{b} H(\mathcal{B}_j)\right) v_{k,j}$$

$$= P_1 \left(I - \frac{2\eta}{b} H(\mathcal{B}_j)\right) (P_1 v_{k,j} + P_{-1} v_{k,j} + P_\perp v_{k,j})$$

$$= P_1 \left(I - \frac{2\eta}{b} H(\mathcal{B}_j)\right) P_1 v_{k,j} + P_1 \left(I - \frac{2\eta}{b} H(\mathcal{B}_j)\right) P_{-1} v_{k,j}$$

$$= \left(I - \frac{2\eta}{b} P_1 H(\mathcal{B}_j) P_1\right) \cdot P_1 v_{k,j} - \left(\frac{2\eta}{b} P_1 H(\mathcal{B}_j) P_{-1}\right) \cdot P_{-1} v_{k,j}.$$

Similarly applying $P_{-1}$ to Eq. (25) we have

$$P_{-1} v_{k,j+1} = P_{-1} \left(I - \frac{2\eta}{b} H(\mathcal{B}_j)\right) v_{k,j}$$

$$= P_{-1} \left(I - \frac{2\eta}{b} H(\mathcal{B}_j)\right) (P_1 v_{k,j} + P_{-1} v_{k,j} + P_\perp v_{k,j})$$

$$= P_{-1} \left(I - \frac{2\eta}{b} H(\mathcal{B}_j)\right) P_1 v_{k,j} + P_{-1} \left(I - \frac{2\eta}{b} H(\mathcal{B}_j)\right) P_{-1} v_{k,j}$$

$$= -\left(\frac{2\eta}{b} P_{-1} H(\mathcal{B}_j) P_1\right) \cdot P_1 v_{k,j} + \left(I - \frac{2\eta}{b} P_{-1} H(\mathcal{B}_j) P_{-1}\right) \cdot P_{-1} v_{k,j}.$$

To sum up we have

$$\begin{pmatrix} P_1 v_{k,j+1} \\ P_{-1} v_{k,j+1} \end{pmatrix} = \begin{pmatrix} I - \frac{2\eta}{b} P_1 H(\mathcal{B}_j) P_1 & -\frac{2\eta}{b} P_1 H(\mathcal{B}_j) P_{-1} \\ -\frac{2\eta}{b} P_{-1} H(\mathcal{B}_j) P_1 & I - \frac{2\eta}{b} P_{-1} H(\mathcal{B}_j) P_{-1} \end{pmatrix} \cdot \begin{pmatrix} P_1 v_{k,j} \\ P_{-1} v_{k,j} \end{pmatrix}$$

Notice that if $1 \notin \mathcal{B}_j$, i.e., $x_1$ is not used in the $j$-th step, then we claim

$$P_1 H(\mathcal{B}_j) = H(\mathcal{B}_j) P_1 = 0,$$

since $H(\mathcal{B}_j) = \sum_{i \in \mathcal{B}_j} x_i x_i^\top$ is composed by the data belonging to the column space of $P_{-1}$. Therefore if $1 \notin \mathcal{B}_j$ we have

$$\begin{pmatrix} P_1 v_{k,j+1} \\ P_{-1} v_{k,j+1} \end{pmatrix} = \begin{pmatrix} I & 0 \\ 0 & I - \frac{2\eta}{b} P_{-1} H(\mathcal{B}_j) P_{-1} \end{pmatrix} \cdot \begin{pmatrix} P_1 v_{k,j} \\ P_{-1} v_{k,j} \end{pmatrix}$$

$\square$

### C.7 PROOF OF LEMMA 10

*Proof of Lemma 10.* Clearly for each component in the production, the column space of $P_1 + P_\perp$, which is $(n - d + 1)$-dimensional, belongs to its eigenspace of eigenvalue 1, which yields the first claim.

In the following, we restrict ourselves in the column space of $P_{-1}$. Let us expand $\mathcal{M}_{-1}$:

$$\begin{aligned} \mathcal{M}_{-1} &= \prod_{j=1}^{m} \left( I - \frac{2\eta}{b} P_{-1} H(\mathcal{B}_j) P_{-1} \right) \\ &= \left( I - \frac{2\eta}{b} P_{-1} H(\mathcal{B}_m) P_{-1} \right) \cdots \left( I - \frac{2\eta}{b} P_{-1} H(\mathcal{B}_1) P_{-1} \right) \\ &= I - \frac{2\eta}{b} \underbrace{\sum_{j=1}^{m} P_{-1} H(\mathcal{B}_j) P_{-1}}_{H_{-1}} \\ &\quad + \left( \frac{2\eta}{b} \right)^2 \underbrace{\sum_{1 \le i < j \le n} P_{-1} H(\mathcal{B}_j) P_{-1} H(\mathcal{B}_i) P_{-1} + \dots}_{C}. \end{aligned}$$

We first analyze matrix $H_{-1}$. Since $H(\mathcal{B}_j) = \sum_{i \in \mathcal{B}_j} x_i x_i^\top$ and $\pi = \{\mathcal{B}_1, \dots, \mathcal{B}_m\}$ is a partition for index set $[n]$, we have

$$\begin{aligned} H_{-1} &= \sum_{j=1}^{m} P_{-1} H(\mathcal{B}_j) P_{-1} \\ &= \sum_{j=1}^{m} P_{-1} \sum_{i \in \mathcal{B}_j} x_i x_i^\top P_{-1} \\ &= P_{-1} \sum_{i=1}^{n} x_i x_i^\top P_{-1} \\ &= P_{-1} X X^\top P_{-1}, \end{aligned}$$

which is exactly the matrix we studied in Lemma 5, and from where we have $H_{-1}$ has eigenvalue zero (with multiplicity being $n - d + 1$) in the column space of $P_1 + P_\perp$, and restricted in the column space of $P_{-1}$, the eigenvalues of $H_{-1}$ belong to $(\lambda_n - n\iota, \lambda_2 + n\iota)$.

Then we analyze matrix $C$.

$$\begin{aligned} P_{-1} H(\mathcal{B}_j) P_{-1} H(\mathcal{B}_i) P)_{-1} &= \left( P_{-1} \sum_{i' \in \mathcal{B}_i} x_{i'} x_{i'}^\top P_{-1} \right) \left( P_{-1} \sum_{j' \in \mathcal{B}_j} x_{j'} x_{j'}^\top P_{-1} \right) \\ &= \sum_{i' \in \mathcal{B}_i} \sum_{j' \in \mathcal{B}_j} (P_{-1} x_{i'}) \langle P_{-1} x_{i'}, P_{-1} x_{j'} \rangle (P_{-1} x_{j'})^\top. \end{aligned} \quad (26)$$

Remember that $\mathcal{B}_i \cap \mathcal{B}_j = \emptyset$ for $i \neq j$, thus $x_{i'} \neq x_{j'}$ for $i' \in \mathcal{B}_i$ and $j' \in \mathcal{B}_j$. Then from Lemma 1 we have,

$$|\langle P_{-1} x_{i'}, P_{-1} x_{j'} \rangle| \le |\langle x_{i'}, x_{j'} \rangle| \le \sqrt{\lambda_{i'} \lambda_{j'}} \cdot \iota \le \iota.$$

Inserting this into Eq. (26) we obtain

$$\|P_{-1}H(\mathcal{B}_j)P_{-1}H(\mathcal{B}_i)P)_{-1}\|_F \le b^2 \cdot \max |\langle P_{-1}x_{i'}, P_{-1}x_{j'}\rangle|^2 \le b^2\iota^2.$$

We can bound the Frobenius norm of the higher degree terms in matrix $C$ in a similar manner; in sum for the Frobenius norm of $C$, we have

$$
\begin{aligned}
\|C\|_F &\le \sum_{s=2}^{m} \left(\frac{2\eta}{b}\right)^s \cdot b^s \cdot \iota^s \cdot \binom{m}{s} \\
&= \sum_{s=2}^{m} (2\eta\iota)^s \cdot \binom{n}{s} \\
&= \sum_{s=0}^{m} (2\eta\iota)^s \cdot \binom{n}{s} - 1 - 2m\eta\iota \\
&= (1 + 2\eta\iota)^m - 1 - 2m\eta\iota \\
&\le 1 + m \cdot 2\eta\iota + \frac{m^2\sqrt{e}}{2} \cdot (2\eta\iota)^2 - 1 - 2m\eta\iota \qquad \text{(for } 2\eta\iota < \frac{1}{2m}\text{)} \\
&\le 4m^2\eta^2\iota^2,
\end{aligned}
$$

where for the second to the last inequality we notice that for $f(t) = (1+t)^m$ and $t \in [0, \frac{1}{2n}]$, we have $f''(t) = m(m-1)(1+t)^{m-2} \le m(m-1)(1+\frac{1}{2m})^{m-2} \le m(m-1) \cdot \sqrt{e}$, which implies $f(t)$ is $(m^2\sqrt{e})$-smooth for $t \in [0, \frac{1}{2m}]$; moreover, by the assumption that $3n\iota < \lambda_n$ and $\eta < \frac{b}{\lambda_n+3n\iota}$, we can indeed verify that

$$2\eta\iota < \frac{2b\iota}{\lambda_n + 3n\iota} \le \frac{2b\iota}{6n\iota} \le \frac{1}{2m}. \tag{27}$$

Now we rephrase $\mathcal{M}_{-1}^\top\mathcal{M}_{-1}$ as

$$
\begin{aligned}
\mathcal{M}_{-1}^\top\mathcal{M}_{-1} &= \left(I - \frac{2\eta}{b}H_{-1} + C^\top\right) \cdot \left(I - \frac{2\eta}{b}H_{-1} + C\right) \\
&= \left(I - \frac{2\eta}{b}H_{-1}\right)^2 + \underbrace{C^\top\left(I - \frac{2\eta}{b}H_{-1}\right) + \left(I - \frac{2\eta}{b}H_{-1}\right)C + C^\top C}_{D}. \tag{28}
\end{aligned}
$$

Restricting ourselves in the column space of $P_{-1}$, the eigenvalues of $H_{-1}$ belong to $(\lambda_n - n\iota, \lambda_2 + n\iota)$, thus the eigenvalues of $\left(I - \frac{2\eta}{b}H_{-1}\right)^2$ are upper bounded by

$$\max\left\{\left(1 - \frac{2\eta}{b}(\lambda_2 + n\iota)\right)^2, \ \left(1 - \frac{2\eta}{b}(\lambda_n - n\iota)\right)^2\right\} < 1, \tag{29}$$

where the last inequality is guaranteed by our assumptions on $\eta$ and $\iota$. For simplicity we defer the verification to the end of the proof.

Consider the following eigen decomposition $I - 2\eta H_{-1} = U \operatorname{diag}(\mu_1, \dots, \mu_{n-1}, 1, \dots, 1) U^\top$, where $\mu_1, \dots, \mu_{n-1} \in (-1, 1)$ by Eq. (29). Then we have

$$
\begin{aligned}
\|(I - \eta H_{-1})C\|_F &= \left\|\operatorname{diag}(\mu_1, \dots, \mu_{n-1}, 1, \dots, 1) U^\top C U\right\|_F \\
&\le \left\|U^\top C U\right\|_F = \|C\|_F.
\end{aligned}
$$

Therefore we can bound the Frobenius norm of $D$ by

$$
\begin{aligned}
\|D\|_F &\le 2\|(I - 2\eta H_{-1})C\|_F + \|C\|_F^2 \\
&\le 2\|C\|_F + \|C\|_F^2 \\
&\le 8m^2\eta^2\iota^2 + 16m^4\eta^4\iota^4 \\
&\le 9m^2\eta^2\iota^2, \tag{30}
\end{aligned}
$$

where the last inequality follows from $2\eta\iota \le 1/(2m)$ proved in Eq. (27).

Finally, applying Hoffman-Wielandt theorem with Eq. (28), (29) and (30), we conclude that, restricted in the column space of $P_{-1}$, the eigenvalues of $\mathcal{M}_{-1}^\top \mathcal{M}_{-1}$ are upper bounded by

$$\max\left\{\left(1 - \frac{2\eta}{b}(\lambda_2 + n\iota)\right)^2 + 9m^2\eta^2\iota^2, \; \left(1 - \frac{2\eta}{b}(\lambda_n - n\iota)\right)^2 + 9m^2\eta^2\iota^2\right\}$$

$$\le \max\left\{\left(\left|1 - \frac{2\eta}{b}(\lambda_2 + n\iota)\right| + 3m\eta\iota\right)^2, \; \left(\left|1 - \frac{2\eta}{b}(\lambda_n - n\iota)\right| + 3m\eta\iota\right)^2\right\}$$

$$:= (q_{-1}(\eta))^2. \tag{31}$$

At this point we left to verify Eq. (29) and

$$q_{-1}(\eta) := \max\left\{\left|1 - \frac{2\eta}{b}(\lambda_2 + n\iota)\right| + \frac{3n\eta\iota}{b}, \; \left|1 - \frac{2\eta}{b}(\lambda_n - n\iota)\right| + \frac{3n\eta\iota}{b}\right\} < 1. \tag{32}$$

Clearly it suffices to verify Eq. (32).

$$\left|1 - \frac{2\eta}{b}(\lambda_2 + n\iota)\right| + \frac{3n\eta\iota}{b} < 1$$

$$\Leftrightarrow \quad \frac{3n\iota}{b}\eta - 1 < 1 - \frac{2(\lambda_2 + n\iota)}{b}\eta < 1 - \frac{3n\iota}{b}\eta$$

$$\Leftrightarrow \quad \begin{cases} \frac{2\lambda_2 - n\iota}{b}\eta > 0 \\ \frac{2\lambda_2 + 5n\iota}{b}\eta < 2 \end{cases}$$

$$\Leftarrow \quad \begin{cases} \eta > 0 \\ 2\lambda_2 - n\iota > 0 \\ \eta < \frac{b}{\lambda_2 + 2.5n\iota} \end{cases}$$

$$\Leftarrow \quad \begin{cases} 3n\iota < \lambda_n \quad (\text{since } \lambda_2 \ge \lambda_n) \\ 0 < \eta < \frac{b}{\lambda_2 + 3n\iota} \end{cases}$$

Similarly, we verify that

$$\left|1 - \frac{2\eta}{b}(\lambda_n - n\iota)\right| + \frac{3n\eta\iota}{b} < 1$$

$$\Leftrightarrow \quad \frac{3n\iota}{b}\eta - 1 < 1 - \frac{2(\lambda_n - n\iota)}{b}\eta < 1 - \frac{3n\iota}{b}\eta$$

$$\Leftrightarrow \quad \begin{cases} \frac{2\lambda_n - 5n\iota}{b}\eta > 0 \\ \frac{2\lambda_n + n\iota}{b}\eta < 2 \end{cases}$$

$$\Leftarrow \quad \begin{cases} \eta > 0 \\ 2\lambda_n - 5n\iota > 0 \\ \eta < \frac{b}{\lambda_n + 0.5n\iota} \end{cases}$$

$$\Leftarrow \quad \begin{cases} 3n\iota < \lambda_n \\ 0 < \eta < \frac{b}{\lambda_2 + 3n\iota} \quad (\text{since } \lambda_2 \ge \lambda_n) \end{cases}$$

These complete our proof. $\qquad\qquad\qquad\qquad\qquad\qquad\qquad\qquad\qquad\qquad\qquad\square$

## C.8 PROOF OF LEMMA 11

*Proof of Lemma 11.* Note that during one epoch of SGD updates, $x_1$ is used for only once. Without loss of generality, assume SGD uses $x_1$ at the $l$-th step, i.e., $1 \in \mathcal{B}_l$ and $1 \notin B_j$ for $j \ne l$. Recursively

applying Lemma 9, we have

$$
\begin{pmatrix} P_1 v_{k,m+1} \\ P_{-1} v_{k,m+1} \end{pmatrix} = \begin{pmatrix} I & 0 \\ 0 & \prod_{j=l+1}^{m} \left(I - \frac{2\eta}{b} P_{-1} H(\mathcal{B}_j) P_{-1}\right) \end{pmatrix} \times
$$
$$
\begin{pmatrix} I - \frac{2\eta}{b} P_1 H(\mathcal{B}_l) P_1 & -\frac{2\eta}{b} P_1 H(\mathcal{B}_l) P_{-1} \\ -\frac{2\eta}{b} P_{-1} H(\mathcal{B}_l) P_1 & I - \frac{2\eta}{b} P_{-1} H(\mathcal{B}_j) P_{-1} \end{pmatrix} \times
$$
$$
\begin{pmatrix} I & 0 \\ 0 & \prod_{j=1}^{l-1} \left(I - \frac{2\eta}{b} P_{-1} H(\mathcal{B}_j) P_{-1}\right) \end{pmatrix} \times \begin{pmatrix} P_1 v_{k,1} \\ P_{-1} v_{k,1} \end{pmatrix}
$$

Let $v_{k+1} = v_{k,m+1}$, $v_k = v_{k,1}$ and

$$
\mathcal{M}_l := I - \frac{2\eta}{b} P_{-1} H(\mathcal{B}_l) P_{-1}
$$
$$
\mathcal{M}_{>l} := \prod_{j=l+1}^{m} \left(I - \frac{2\eta}{b} P_{-1} H(\mathcal{B}_j) P_{-1}\right)
$$
$$
\mathcal{M}_{<l} := \prod_{j=1}^{l-1} \left(I - \frac{2\eta}{b} P_{-1} H(\mathcal{B}_j) P_{-1}\right)
$$
$$
\mathcal{M}_{-1} := \mathcal{M}_{>l} \cdot \mathcal{M}_l \cdot \mathcal{M}_{<l} = \prod_{j=1}^{m} \left(I - \frac{2\eta}{b} P_{-1} H(\mathcal{B}_j) P_{-1}\right)
$$

then we have

$$
\begin{pmatrix} P_1 v_{k+1} \\ P_{-1} v_{k+1} \end{pmatrix} = \begin{pmatrix} I & 0 \\ 0 & \mathcal{M}_{>l} \end{pmatrix} \begin{pmatrix} I - \frac{2\eta}{b} P_1 H(\mathcal{B}_l) P_1 & -\frac{2\eta}{b} P_1 H(\mathcal{B}_l) P_{-1} \\ -\frac{2\eta}{b} P_{-1} H(\mathcal{B}_l) P_1 & \mathcal{M}_l \end{pmatrix} \begin{pmatrix} I & 0 \\ 0 & \mathcal{M}_{<l} \end{pmatrix} \begin{pmatrix} P_1 v_k \\ P_{-1} v_k \end{pmatrix}
$$
$$
= \begin{pmatrix} I - \frac{2\eta}{b} P_1 H(\mathcal{B}_l) P_1 & -\left(\frac{2\eta}{b} P_1 H(\mathcal{B}_l) P_{-1}\right) \mathcal{M}_{<l} \\ -\mathcal{M}_{>l} \left(\frac{2\eta}{b} P_{-1} H(\mathcal{B}_l) P_1\right) & \mathcal{M}_{-1} \end{pmatrix} \begin{pmatrix} P_1 v_k \\ P_{-1} v_k \end{pmatrix} \tag{33}
$$

In the following we bound the norm of each entries in the above coefficient matrix.

According to Lemma 5, we have the eigenvalues of $P_{-1} H(\mathcal{B}_j) P_{-1}$ are upper bounded by $\lambda_2 + n\iota$. Thus the assumption $\eta < \frac{b}{\lambda_2 + 2n\iota}$ yields

$$
\left\| I - \frac{2\eta}{b} P_{-1} H(\mathcal{B}_j) P_{-1} \right\|_2 \leq 1,
$$

which further yields

$$
\|\mathcal{M}_{>l}\|_2 \leq 1, \quad \|\mathcal{M}_{<l}\|_2 \leq 1. \tag{34}
$$

On the other hand notice that $P_1 x_i = 0$ for $i \neq 1$, thus

$$
P_1 H(\mathcal{B}_l) P_{-1} = P_1 \sum_{i \in \mathcal{B}_l} x_i x_i^\top P_{-1} = P_1 x_1 x_1^\top P_{-1},
$$
$$
P_{-1} H(\mathcal{B}_l) P_1 = P_{-1} \sum_{i \in \mathcal{B}_l} x_i x_i^\top P_1 = P_{-1} x_1 x_1^\top P_1,
$$

which yield

$$
\max\left\{ \|P_1 H(\mathcal{B}_l) P_{-1}\|_2, \|P_{-1} H(\mathcal{B}_l) P_1\|_2 \right\} \leq \|P_1 x_1\|_2 \cdot \|P_{-1} x_1\|_2 \leq 2\sqrt{n}\iota, \tag{35}
$$

where the last inequality is from Lemma 3 and $\lambda_1 = \|x_1\|^2 \leq 1$. Eq. (34) and (35) imply

$$
\max\left\{ \left\| \left(\frac{2\eta}{b} P_1 H(\mathcal{B}_l) P_{-1}\right) \mathcal{M}_{<l} \right\|_2, \left\| \mathcal{M}_{>l} \left(\frac{2\eta}{b} P_{-1} H(\mathcal{B}_l) P_1\right) \right\|_2 \right\} \leq \frac{4\eta\sqrt{n}\iota}{b} =: \xi(\eta) \tag{36}
$$

Next, by $P_1 x_i = 0$ for $i \neq 1$ we have

$$
P_1 H(\mathcal{B}_l) P_1 = P_1 \sum_{i \in \mathcal{B}_l} x_i x_i^\top P_1 = P_1 x_1 x_1^\top P_1 = (P_1 x_1)(P_1 x_1)^\top,
$$

from where we know $\|P_1 x_1\|_2^2$ is the only non-zero eigenvalue of the rank-1 matrix $P_1 H(\mathcal{B}_l) P_1$, and the corresponding eigenspace is the column space of $P_1$. Therefore $1 - \frac{2\eta}{b} \|P_1 x_1\|_2^2$ is an eigenvalue of the matrix $I - \frac{2\eta}{b} P_1 H(\mathcal{B}_l) P_1$, and the corresponding eigenspace is the column space of $P_1$, which implies

$$
\begin{aligned}
\left\| \left( I - \frac{2\eta}{b} P_1 H(\mathcal{B}_l) P_1 \right) P_1 v_k \right\|_2 &= \left\| \left( 1 - \frac{2\eta}{b} \|P_1 x_1\|_2^2 \right) P_1 v_k \right\| \\
&= \left| 1 - \frac{2\eta}{b} \|P_1 x_1\|_2^2 \right| \cdot \|P_1 v_k\|_2 \\
&=: q_1(\eta) \cdot \|P_1 v_k\|_2 .
\end{aligned}
\tag{37}
$$

Finally, according to Lemma 10, we have, restricted in the column space of $P_{-1}$, the right eigenvalues of $\mathcal{M}_{-1}$ is upper bounded by $(q_{-1}(\eta))^2$, which implies

$$
\|\mathcal{M}_{-1} P_{-1} v_k\|_2 \le q_{-1}(\eta) \cdot \|P_{-1} v_k\|_2 .
\tag{38}
$$

Note we have $q_{-1}(\eta) < 1$ by Lemma 10.

Combining Eq. (33) with Eq. (36), (37), (38), and letting $B_k := \|P_1 v_k\|_2$, $A_k := \|P_{-1} v_k\|_2$, we obtain

$$
\begin{aligned}
B_{k+1} &\le q_1(\eta) \cdot B_k + \xi(\eta) \cdot A_k \\
B_{k+1} &\ge q_1(\eta) \cdot B_k - \xi(\eta) \cdot A_k \\
A_{k+1} &\le q_{-1} \cdot (\eta) A_k + \xi(\eta) \cdot B_k.
\end{aligned}
$$

$\square$

## C.9 Proof of Lemma 12

*Proof of Lemma 12.* Let

$$
\begin{aligned}
\xi := \xi(\eta) &= \frac{4\eta\sqrt{n}\iota}{b}, \\
q_1 := q_1(\eta) &= \left| 1 - \frac{2\eta\lambda_1}{b} \|P_1 \bar{x}_1\|_2^2 \right|, \\
q_{-1} := q_{-1}(\eta) &= \max\left\{ \left| 1 - \frac{2\eta}{b}(\lambda_2 + n\iota) \right| + \frac{3n\eta\iota}{b}, \ \left| 1 - \frac{2\eta}{b}(\lambda_n - n\iota) \right| + \frac{3n\eta\iota}{b} \right\}.
\end{aligned}
\tag{39}
$$

Then for $0 < k \le k_1$, Lemma 11 gives us

$$
B_k \ge q_1 B_{k-1} - \xi A_{k-1},
\tag{40}
$$

$$
\begin{pmatrix} A_k \\ B_k \end{pmatrix} \le \begin{pmatrix} q_{-1} & \xi \\ \xi & q_1 \end{pmatrix} \cdot \begin{pmatrix} A_{k-1} \\ B_{k-1} \end{pmatrix},
\tag{41}
$$

where "$\le$" means "entry-wisely smaller than".

Let $\theta, \rho_{-1}, \rho_1$ determine the eigen decomposition of the coefficient matrix, i.e.,

$$
\begin{pmatrix} q_{-1} & \xi \\ \xi & q_1 \end{pmatrix} = \begin{pmatrix} \cos\theta & \sin\theta \\ -\sin\theta & \cos\theta \end{pmatrix} \begin{pmatrix} \rho_{-1} & 0 \\ 0 & \rho_1 \end{pmatrix} \begin{pmatrix} \cos\theta & -\sin\theta \\ \sin\theta & \cos\theta \end{pmatrix}.
\tag{42}
$$

Then Eq. (41) and Eq. (42) yield

$$
\begin{pmatrix} A_k \\ B_k \end{pmatrix} \leq \begin{pmatrix} q_{-1} & \xi \\ \xi & q_1 \end{pmatrix}^k \cdot \begin{pmatrix} A_0 \\ B_0 \end{pmatrix}
$$
$$
= \begin{pmatrix} \cos\theta & \sin\theta \\ -\sin\theta & \cos\theta \end{pmatrix} \begin{pmatrix} \rho_{-1}^k & 0 \\ 0 & \rho_1^k \end{pmatrix} \begin{pmatrix} \cos\theta & -\sin\theta \\ \sin\theta & \cos\theta \end{pmatrix} \begin{pmatrix} A_0 \\ B_0 \end{pmatrix}
$$
$$
= \begin{pmatrix} \rho_{-1}^k + (\rho_1^k - \rho_{-1}^k)\sin^2\theta & (\rho_1^k - \rho_{-1}^k)\cos\theta\sin\theta \\ (\rho_1^k - \rho_{-1}^k)\cos\theta\sin\theta & \rho_1^k - (\rho_1^k - \rho_{-1}^k)\sin^2\theta \end{pmatrix} \begin{pmatrix} A_0 \\ B_0 \end{pmatrix}
$$
$$
= \begin{pmatrix} A_0 \cdot \rho_{-1}^k + (\rho_1^k - \rho_{-1}^k)(A_0\sin\theta + B_0\cos\theta)\sin\theta \\ B_0 \cdot \rho_1^k + (\rho_1^k - \rho_{-1}^k)(A_0\cos\theta - B_0\sin\theta)\sin\theta \end{pmatrix}
$$
$$
\leq \begin{pmatrix} A_0 \cdot \rho_{-1}^k + \left| \rho_1^k - \rho_{-1}^k \right| \sqrt{A_0^2 + B_0^2}\sin\theta \\ B_0 \cdot \rho_1^k + \left| \rho_1^k - \rho_{-1}^k \right| \sqrt{A_0^2 + B_0^2}\sin\theta \end{pmatrix}
$$
$$
= \begin{pmatrix} A_0 \cdot \rho_{-1}^k + \left| \rho_1^k - \rho_{-1}^k \right| \cdot \|Pv_0\|_2 \cdot \sin\theta \\ B_0 \cdot \rho_1^k + \left| \rho_1^k - \rho_{-1}^k \right| \cdot \|Pv_0\|_2 \cdot \sin\theta \end{pmatrix}. \tag{43}
$$

We claim the following inequalities hold by our assumptions:

$$
0 < \rho_{-1} < 1 < \rho_1 \leq q_1 + \xi \tag{44a}
$$
$$
\rho_{-1}^{k_1} \|Pv_0\|_2 \leq \frac{\epsilon}{2} \cdot \beta \tag{44b}
$$
$$
\rho_1^{k_1} \|Pv_0\|_2 \sin\theta \leq \frac{\epsilon}{2} \cdot \beta, \tag{44c}
$$
$$
\xi \cdot \left( A_0 + \frac{\epsilon\beta_0}{2} \right) < (q_1 - 1)\beta_0. \tag{44d}
$$

The verification of Eq. (44) is left later. In the following we prove the conclusions using Eq. (44).

We first bound $A_{k_1}$ using Eq. (43) and Eq. (44):

$$
A_{k_1} \leq A_0 \cdot \rho_{-1}^{k_1} + \left| \rho_1^{k_1} - \rho_{-1}^{k_1} \right| \cdot \|Pv_0\|_2 \cdot \sin\theta
$$
$$
\leq \|Pv_0\|_2 \cdot \rho_{-1}^{k_1} + \rho_1^{k_1} \cdot \|Pv_0\|_2 \cdot \sin\theta
$$
$$
\leq \frac{\epsilon}{2} \cdot \beta + \frac{\epsilon}{2} \cdot \beta
$$
$$
= \epsilon \cdot \beta,
$$

which justifies the first conclusion. In addition we can obtain an uniform upper bound for $A_k$ for $k = 0, 1, \ldots, k_1$:

$$
A_k \leq A_0 \cdot \rho_{-1}^k + \left| \rho_1^k - \rho_{-1}^k \right| \cdot \|Pv_0\|_2 \cdot \sin\theta
$$
$$
\leq A_0 + \rho_1^k \cdot \|Pv_0\|_2 \cdot \sin\theta
$$
$$
\leq A_0 + \frac{\epsilon}{2} \cdot \beta. \tag{45}
$$

Next we bound $B_{k_1}$ using Eq. (43) and Eq. (44):

$$
B_{k_1} \leq B_0 \cdot \rho_1^{k_1} + \left| \rho_1^{k_1} - \rho_{-1}^{k_1} \right| \cdot \|Pv_0\|_2 \cdot \sin\theta
$$
$$
\leq \|Pv_0\|_2 \cdot \rho_1^{k_1} + \rho_1^{k_1} \cdot \|Pv_0\|_2 \cdot \sin\theta
$$
$$
\leq \|Pv_0\|_2 \cdot \rho_1^{k_1} + \frac{\epsilon}{2} \cdot \beta,
$$

which justifies the second conclusion.

We proceed to derive the uniform lower bound for $B_k$ for $k = 0, 1, \ldots, k_1$. We do it by induction. For $k = 0$, by assumption we have $B_0 \geq \beta_0$. Suppose $B_{k-1} \geq \beta_0$, then by Eq. (40), (44) and (45)

we have

$$
\begin{aligned}
B_k &\geq q_1 \cdot B_{k-1} - \xi \cdot A_{k-1} \\
&\geq q_1 \cdot \beta_0 - \xi \cdot \left( A_0 + \frac{\epsilon}{2}\beta \right) \\
&\geq q_1 \cdot \beta_0 - \xi \cdot \left( A_0 + \frac{\epsilon}{2}\beta_0 \right) \\
&\geq \beta_0,
\end{aligned}
$$

which justifies the third conclusion.

**Verification of Eq. (44)**

From Eq. (42) and Gershgorin circle theorem we have

$$
\begin{aligned}
q_1 - \xi \leq \rho_1 \leq q_1 + \xi, \\
q_{-1} - \xi \leq \rho_{-1} \leq q_{-1} + \xi.
\end{aligned}
\tag{46}
$$

Moreover, reformatting Eq. (42) as

$$
\begin{aligned}
\begin{pmatrix} q_{-1} & \xi \\ \xi & q_1 \end{pmatrix} &= \begin{pmatrix} \cos\theta & \sin\theta \\ -\sin\theta & \cos\theta \end{pmatrix} \begin{pmatrix} \rho_{-1} & 0 \\ 0 & \rho_1 \end{pmatrix} \begin{pmatrix} \cos\theta & -\sin\theta \\ \sin\theta & \cos\theta \end{pmatrix} \\
&= \begin{pmatrix} \rho_{-1}\cos^2\theta + \rho_1\sin^2\theta & (\rho_1 - \rho_{-1})\cos\theta\sin\theta \\ (\rho_1 - \rho_{-1})\cos\theta\sin\theta & \rho_{-1}\sin^2\theta + \rho_1\cos^2\theta \end{pmatrix} \\
&= \begin{pmatrix} \rho_{-1} + (\rho_1 - \rho_{-1})\sin^2\theta & (\rho_1 - \rho_{-1})\cos\theta\sin\theta \\ (\rho_1 - \rho_{-1})\cos\theta\sin\theta & \rho_1 - (\rho_1 - \rho_{-1})\sin^2\theta \end{pmatrix},
\end{aligned}
$$

we then have

$$
\frac{\xi}{q_1 - q_{-1}} = \frac{(\rho_1 - \rho_{-1})\cos\theta\sin\theta}{(\rho_1 - \rho_{-1})(1 - 2\sin^2\theta)} = \frac{1}{2}\tan 2\theta.
\tag{47}
$$

For Eq. (44a), using Eq. (46) it suffices to show

$$
\begin{aligned}
0 &< q_1 - \xi, & \text{(48a)} \\
q_{-1} + \xi &< 1, & \text{(48b)} \\
1 &< q_1 - \xi. & \text{(48c)}
\end{aligned}
$$

Notice the definitions of $q_1$, $q_{-1}$ and $\xi$ are given in Eq. (39). Firstly, Eq. (48c) holds trivially when $\sqrt{n} > 4/3$. Secondly, for Eq. (48b), noticing that $\xi = \frac{4\eta\sqrt{n}\iota}{b} \leq \frac{\eta n \iota}{b}$ when $n \geq 16$, it suffices to show

$$
\max\left\{ \left| 1 - \frac{2\eta}{b}(\lambda_2 + n\iota) \right| + \frac{4n\eta\iota}{b}, \left| 1 - \frac{2\eta}{b}(\lambda_n - n\iota) \right| + \frac{4n\eta\iota}{b} \right\} < 1
$$

$$
\Leftrightarrow \begin{cases} \frac{4n\iota}{b}\eta - 1 < 1 - \frac{2(\lambda_2 + n\iota)}{b}\eta < 1 - \frac{4n\iota}{b}\eta \\ \frac{4n\iota}{b}\eta - 1 < 1 - \frac{2(\lambda_n - n\iota)}{b}\eta < 1 - \frac{4n\iota}{b}\eta \end{cases}
$$

$$
\Leftrightarrow \begin{cases} \frac{2\lambda_2 - 2n\iota}{b}\eta > 0 \\ \frac{2\lambda_2 + 6n\iota}{b}\eta < 2 \\ \frac{2\lambda_n - 6n\iota}{b}\eta > 0 \\ \frac{2\lambda_n + 2n\iota}{b}\eta < 2 \end{cases}
$$

$$
\Leftarrow \begin{cases} \eta > 0 \\ \lambda_2 - n\iota > 0 \\ \lambda_n - 3n\iota > 0 \\ \eta < \frac{b}{\lambda_2 + 3n\iota} \\ \eta < \frac{b}{\lambda_n + n\iota} \end{cases}
$$

$$
\Leftarrow \begin{cases} 3n\iota < \lambda_n \\ 0 < \eta < \frac{b}{\lambda_2 + 3n\iota} \end{cases}
$$

which are given in assumptions. Thirdly, for Eq. (48c) it suffices to show

$$\frac{2\eta\lambda_1}{b}\|P_1\bar{x}_1\|_2^2 - 1 - \frac{4\eta\sqrt{n}\iota}{b} > 1$$

$$\Leftarrow \quad \frac{2\lambda_1(1-4n\iota^2)}{b}\eta - \frac{4\sqrt{n}\iota}{b}\eta > 2 \qquad \text{(by Lemma 3)}$$

$$\Leftarrow \quad \eta > \frac{b}{\lambda_1(1-4n\iota^2) - 2\sqrt{n}\iota}$$

$$\Leftarrow \quad \eta > \frac{b}{\lambda_1 - 3\sqrt{n}\iota}, \qquad \text{(since } n\iota < 1\text{)}$$

which are given in assumptions.

For Eq. (44b), it suffices to show set

$$k_1 = 1 + \frac{\log\frac{0.5\epsilon\beta}{\|Pv_0\|_2}}{\log\rho_{-1}} = \mathcal{O}\left(\log\frac{1}{\epsilon\beta}\right),$$

as given in assumptions.

For Eq. (44c), using Eq. (44b) it suffices to show

$$\sin\theta \le \left(\frac{\rho_{-1}}{\rho_1}\right)^{k_1} = \frac{\rho_{-1}}{\rho_1} \cdot \left(\frac{0.5\epsilon\beta}{\|Pv_0\|_2}\right)^{1 - \frac{\log\rho_1}{\log\rho_{-1}}}$$

$$\Leftarrow \quad \sin\theta \le \frac{q_{-1} - \xi}{q_1 + \xi} \cdot \left(\frac{0.5\epsilon\beta}{\|Pv_0\|_2}\right)^{1 - \frac{\log(q_1+\xi)}{\log(q_{-1}-\xi)}}$$

$$\Leftarrow \quad \xi \le 0.9\left(q_1 - q_{-1}\right) \cdot \frac{q_{-1} - \xi}{q_1 + \xi} \cdot \left(\frac{0.5\epsilon\beta}{\|Pv_0\|_2}\right)^{1 - \frac{\log(q_1+\xi)}{\log(q_{-1}-\xi)}} \qquad \text{(by Eq. (47))}$$

$$\Leftarrow \quad \sqrt{n}\iota \le \text{poly}\left(\epsilon\beta\right). \qquad \text{(by Eq. (39))}$$

For Eq. (44c), it suffices to show

$$\xi \le \frac{(q_1 - 1)\beta_0}{A_0 + 0.5\epsilon\beta_0}$$

$$\Leftarrow \quad \sqrt{n}\iota \le \mathcal{O}\left(1\right). \qquad \text{(by Eq. (39))}$$

$\square$

## C.10   PROOF OF LEMMA 13

*Proof of Lemma 13.* Let

$$\xi' := \xi(\eta') = \frac{4\eta'\sqrt{n}\iota}{b},$$

$$q_1' := q_1(\eta') = \left|1 - \frac{2\eta'\lambda_1}{b}\|P_1\bar{x}_1\|_2^2\right|, \qquad (49)$$

$$q_{-1}' := q_{-1}(\eta') = \max\left\{\left|1 - \frac{2\eta'}{b}(\lambda_2 + n\iota)\right| + \frac{3n\eta'\iota}{b}, \ \left|1 - \frac{2\eta'}{b}(\lambda_n - n\iota)\right| + \frac{3n\eta'\iota}{b}\right\}.$$

Then for $k_1 < k \le k_2$, Lemma 11 gives us

$$\begin{pmatrix} A_k \\ B_k \end{pmatrix} \le \begin{pmatrix} q_{-1}' & \xi' \\ \xi' & q_1' \end{pmatrix} \cdot \begin{pmatrix} A_{k-1} \\ B_{k-1} \end{pmatrix}, \qquad (50)$$

where "$\le$" means "entry-wisely smaller than". Denote

$$B := \|Pv_0\|_2 \cdot \rho_1^{k_1} + \frac{\epsilon}{2} \cdot \beta = \text{poly}\left(\frac{1}{\epsilon\beta}\right). \qquad (51)$$

We claim the following inequalities hold by our assumptions:

$$0 < q'_1 < q'_{-1} < 1, \tag{52a}$$

$$\xi' \cdot \epsilon \le q'_{-1} - q'_1, \tag{52b}$$

$$\xi' \cdot B \le (1 - q'_{-1}) \cdot \epsilon \cdot \beta. \tag{52c}$$

The verification of Eq. (52) is left later. In the following we prove the main conclusions in the lemma using Eq. (52). We proceed by induction. Clearly the conclusions are true for $k = k_1$. Suppose for $k_1, \ldots, k - 1$, the conclusions are also true. Then the induction assumptions give us

$$A_{k-1} \le \epsilon \cdot \beta, \tag{53}$$

$$B_{k-1} \le B_{k_1} \le B, \tag{54}$$

where the last inequality is due to $B \ge B_{k_1} \ge \beta_0 > \beta$. Then by Eq. (50) we have

$$
\begin{aligned}
B_k &\le q'_1 \cdot B_{k-1} + \xi' \cdot A_{k-1} \\
&\le q'_1 \cdot B_{k-1} + \xi' \cdot \epsilon \cdot \beta && \text{(by Eq. (53))} \\
&\le q'_1 \cdot B_{k-1} + (q'_{-1} - q'_1) \cdot \beta && \text{(by Eq. (52b))} \\
&\le \begin{cases} q'_{-1} \cdot B_{k-1}, & B_{k-1} > \beta, \\ \beta, & B_{k-1} \le \beta. \end{cases} && \text{(by Eq. (52a))}
\end{aligned}
$$

Also by Eq. (50) we have

$$
\begin{aligned}
A_k &\le q'_{-1} \cdot A_{k-1} + \xi' \cdot B_k \\
&\le q'_{-1} \cdot \epsilon \cdot \beta + \xi' \cdot B && \text{(by Eq. (53) and (54))} \\
&\le q'_{-1} \cdot \epsilon \cdot \beta + (1 - q'_{-1}) \cdot \epsilon \cdot \beta && \text{(by Eq. (52c))} \\
&= \epsilon \cdot \beta.
\end{aligned}
$$

**Verification of Eq. (52)**    Notice the definitions of $q'_1$, $q'_{-1}$ and $\xi'$ are given in Eq. (39). Recall $q'_{-1} < 1$ is already justified by the choice of learning rate $\eta' < \frac{1}{\lambda_2 + 3n\iota}$ (e.g., see Lemma 10), thus for Eq. (52a), it suffices to show

$$
0 < 1 - \frac{2\eta'\lambda_1}{b} \|P_1 \bar{x}_1\|_2^2 < 1 - \frac{2\eta'}{b}(\lambda_n - n\iota) + \frac{3n\eta'\iota}{b}
$$

$$
\Leftarrow \quad \begin{cases} 1 - \frac{2\eta'\lambda_1}{b} > 0 \\ 2\lambda_1(1 - 4n\iota^2) > 2(\lambda_n - n\iota) + 3n\iota \end{cases} \quad \text{(by Lemma 3)}
$$

$$
\Leftarrow \quad \begin{cases} \eta' < \frac{b}{2\lambda_1} \\ \lambda_1 > \lambda_n + 2n\iota \end{cases}
$$

which are given in assumptions.

For Eq. (52b), it suffices to show

$$
\xi' \le \frac{1}{\epsilon} \cdot \left( q'_{-1} - q'_1 \right)
$$

$$
\Leftarrow \quad \sqrt{n}\iota \le \mathcal{O}\left( \frac{1}{\epsilon} \right),
$$

which is implied by $n\iota \le \text{poly}(\epsilon\beta)$.

For Eq. (52c), it suffices to show

$$
\xi' \le \frac{1}{B} \cdot \epsilon \cdot \left( 1 - q'_{-1} \right) \beta
$$

$$
\Leftarrow \quad \sqrt{n}\iota \le \text{poly}(\epsilon\beta). \quad \text{(by Eq. (51))}
$$

We complete our proof. $\qquad\square$

## C.11    PROOF OF LEMMA 14

*Proof of Lemma 14.* For the empirical loss,

$$L_{\mathcal{S}}(u) = \frac{1}{n}(w - w_*)^\top XX^\top (w - w_*) = \frac{1}{n}u^\top G^\top XX^\top Gu = \frac{1}{n}u^\top \Gamma u = \frac{1}{n}\sum_{i=1}^{n}\gamma_i\left(u^{(i)}\right)^2.$$

For the population loss,

$$L_{\mathcal{D}}(u) = \mu\|w - w_*\|_2^2 = \mu\|Gu\|_2^2 = \mu\|u\|_2^2.$$

For the hypothesis class $\mathcal{H}_{\mathcal{S}} = \{w \in \mathbb{R}^d : P_\perp w = P_\perp w_0\}$, Note $P_\perp G = \text{diag}(0, \ldots, 0, 1, \ldots, 1)$. Apply $w - w_* = Gu$ and notice $w_0 - w_* = Gu_0$, then we obtain

$$\mathcal{H}_{\mathcal{S}} = \{u \in \mathbb{R}^d : P_\perp Gu = P_\perp Gu_0\}$$
$$= \{u \in \mathbb{R}^d : u^{(i)} = u_0^{(i)}, \text{ for } i = n+1, \ldots, d\}.$$

For the level set, we only need to note that $L_{\mathcal{S}}^* = \inf_{u \in \mathcal{H}_{\mathcal{S}}} L_{\mathcal{S}}(u) = 0$.

As for the estimation error, we note that

$$\inf_{u \in \mathcal{U}} L_{\mathcal{D}}(u) = \mu\sum_{i=n+1}^{d}\left(u_0^{(i)}\right)^2,$$

thus for $u \in \mathcal{U}$, we have

$$\Delta(u) = L(u) - \inf_{u' \in \mathcal{U}} L(u') = \mu\|u\|_2^2 - \mu\sum_{i=n+1}^{d}\left(u_0^{(i)}\right)^2$$

$$= \mu\sum_{i=1}^{n}\left(u^{(i)}\right)^2 + \mu\sum_{i=n+1}^{d}\left(u^{(i)}\right)^2 - \mu\sum_{i=n+1}^{d}\left(u_0^{(i)}\right)^2$$

$$= \mu\sum_{i=1}^{n}\left(u^{(i)}\right)^2.$$

Now consider $u \in \mathcal{U}$, i.e., $\frac{1}{n}\sum_{i=1}^{n}\gamma_i\left(u^{(i)}\right)^2 = \alpha$, then

$$\Delta_* = \inf_{u \in \mathcal{U}} \Delta(u) = \inf_{n\alpha = \sum_{i=1}^{n}\gamma_i\left(u^{(i)}\right)^2} \mu\sum_{i=1}^{n}\left(u^{(i)}\right)^2 = \frac{\mu n\alpha}{\gamma_1},$$

where the inferior is attended when, e.g., $u^{(1)} = \pm\sqrt{\frac{n\alpha}{\gamma_1}}$ and $u^{(2)} = \cdots = u^{(n)} = 0$. $\qquad\square$

## D    DETAILS OF THE EXPERIMENTS

In this section, we describe the details for our experiments.

### D.1    2-D EXAMPLE

This part corresponds to Section 3 and Figure 1.

The two training data points are

$$x_1 = (\sqrt{\kappa},\ 0)^\top,\quad x_2 = (0,\ 1)^\top,\quad \kappa = 4,$$

and the corresponding individual losses are

$$\ell_1(w) = w^\top x_1 x_1^\top w,\quad \ell_2(w) = w^\top x_2 x_2^\top w.$$

Then the training loss is

$$L_{\mathcal{S}}(w) = 0.5(\ell_1(w) + \ell_2(w)).$$

We initialize the algorithms from $w_0 = (0.6, 0.6)^\top$. Two kinds of learning rate regime are considered. In the small learning rate regime, the learning rate is

$$\eta_k = 0.1/\kappa, \quad k = 1, \dots, 800.$$

In the moderate learning rate regime, the learning rate is

$$\eta_k = \begin{cases} 1.1/\kappa, & k = 1, \dots, 100; \\ 0.1/\kappa, & k = 101, \dots, 800. \end{cases}$$

For SGD, the mini-batch size is $1$.

### D.2 LINEAR REGRESSION ON SYNTHETIC DATA

This part corresponds to Section 4 and Figure 2(a).

The model is an overparameterized linear model, with $d = 10^4$ and $n = 100$. The true parameter $w_*$ is randomly drawn from an $d$-dimensional Gaussian distribution, $\mathcal{N}(0, 0.1 \cdot I_{d \times d})$.

We then randomly draw $n = 100$ samples from the $d$-dimensional space as described in Section 4, where $\zeta \sim \mathcal{U}([0.5, 1])$.

We initialize the algorithms from zero. We consider two kinds of learning rate regimes. The small learning rate scheme is specified by

$$\eta_k = 0.2, \quad k = 1, \dots, 10^4.$$

The moderate learning rate scheme is specified by

$$\eta_k = \begin{cases} 1.05, & k = 1, \dots, 2 \times 10^3; \\ 0.1, & k = 1 + 2 \times 10^3, \dots, 3 \times 10^3. \end{cases}$$

For SGD, the mini-batch size is $1$.

### D.3 NEURAL NETWORK ON A SUBSET OF FASHIONMNIST

This part corresponds to Figure 2(b) and Figure 3.

**Model** We use a LeNet-alike convolutional network:

$$\text{input} \Rightarrow \text{conv1} \Rightarrow \text{ReLU} \Rightarrow \text{max\_pool} \Rightarrow \text{conv2} \Rightarrow \text{ReLU} \Rightarrow$$
$$\text{max\_pool} \Rightarrow \text{fc1} \Rightarrow \text{ReLU} \Rightarrow \text{fc2} \Rightarrow \text{ReLU} \Rightarrow \text{linear} \Rightarrow \text{output}.$$

The first convolutional layer uses $5 \times 5$ kernels with $10$ channels and no padding and the second convolutional layer uses $5 \times 5$ kernels with $16$ channels and no padding. The number of hidden units between the two fully connected layers are $60$.

**Dataset** https://github.com/zalandoresearch/fashion-mnist

We randomly choose $2,000$ original test data as our training set, and use the $60,000$ original training data as our test set. Thus we have $2,000$ training data and $60,000$ test data. We scale the image data to $[0, 1]$.

**Algorithms** We randomly initialize the algorithms from a Gaussian distribution with zero mean and standard deviation $0.02$. We consider two kinds of learning rate regimes. The small learning rate scheme is specified by

$$\eta_k = 10^{-3}, \quad k = 1, \dots, 10^4.$$

The moderate learning rate scheme is specified by

$$\eta_k = \begin{cases} 10^{-2}, & k = 1, \dots, 2.5 \times 10^3; \\ 10^{-3}, & k = 1 + 2.5 \times 10^3, \dots, 10^4. \end{cases}$$

For SGD, the mini-batch size is $25$. For both GD and SGD, the weight decay parameter is set as $0.002$.

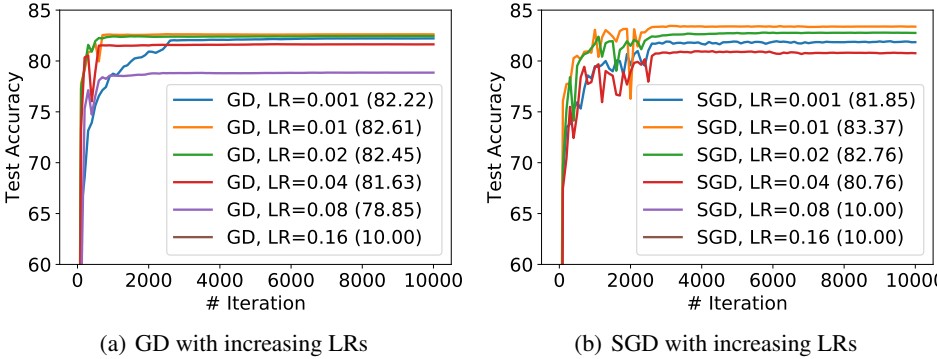

(a) GD with increasing LRs   (b) SGD with increasing LRs

Figure 4: Test accuracy vs. number of iteration for algorithms with increasing learning rates. The model is a 5-layer convolutional neural network, and the dataset is a subset of FashionMNIST dataset. Details are described in Appendix D.3.

Table 1: Test accuracy $(\%)$ of GD/SGD in different LR on a neural network example

| Experiment | #1 | #2 | #3 | #4 | #5 | #6 | #7 | #8 | #9 | #10 |
|---|---|---|---|---|---|---|---|---|---|---|
| SGD small LR | 81.57 | 81.85 | 81.71 | 81.74 | 82.29 | 81.38 | 82.10 | 82.05 | 81.95 | 81.77 |
| GD small LR | 81.45 | 81.25 | 82.22 | 81.73 | 81.89 | 81.32 | 81.71 | 81.75 | 81.43 | 81.35 |
| SGD moderate LR | 82.32 | 83.37 | 81.40 | 82.30 | 82.58 | 82.61 | 81.68 | 83.24 | 82.53 | 82.65 |
| GD moderate LR | 81.54 | 82.62 | 78.39 | 81.81 | 81.86 | 80.91 | 81.84 | 82.15 | 81.65 | 81.83 |

**Relative Rayleigh quotient** We discuss the *relative Rayleigh quotients* calculated in Figure 2(b). Unlike the linear regression model where the Hessian is a constant, for neural networks the loss function is non-convex. In other words, not only there are multiple local minima, but also the Hessian varies at different points. Therefore, a direct comparison in terms of the Rayleigh quotients for the iterates of different algorithms makes little sense. Instead, we consider the relative Rayleigh quotient, i.e., the Rayleigh quotient normalized by the maximum absolute eigenvalue of the Hessian at that point. Mathematically, the relative Rayleigh quotient is defined as

$$\mathrm{RRQ}(w) := \frac{\frac{\nabla L(w)^\top}{\|\nabla L(w)\|_2} \cdot \nabla^2 L(w) \cdot \frac{\nabla L(w)}{\|\nabla L(w)\|_2}}{\|\nabla^2 L(w)\|_2},$$

where $\nabla L(w)/\|\nabla L(w)\|_2$ is the convergence direction of gradient methods, and $\|\nabla^2 L(w)\|_2$ is the operator norm of the Hessian, i.e., its maximum absolute eigenvalue. Note that it is computationally hard in practice to project a parameter onto the data manifold, thus we use the vanilla convergence direction instead of the projected one in the above definition of the relative Rayleigh quotient. Since our goal is to compare the relative Rayleigh quotient between different algorithms, this simplification would not affect our conclusions. We obtain the maximum absolute eigenvalue of the Hessian by running power method for $5$ iterates.

**Additional experiments: GD and SGD with increasing learning rates** We conduct numerical experiments of training neural networks on a subset of FashionMNIST dataset for SGD and GD with different learning rates $\eta \in \{0.001, 0.01, 0.02, 0.04, 0.08, 0.16\}$. The test accuracy results are displayed in Figure 4.

Several conclusions can be drawn from the plots. (1) For a learning rate over $\eta = 0.08$, SGD cannot converge (thus only gives about $10\%$ test accuracy). SGD generalizes best with a moderate learning rate $\eta = 0.01$. (2) GD fails to converge when $\eta = 0.16$. Also, as the learning rate increases, the test accuracy of GD first increases then decreases; but even at its peak ($\eta = 0.01$), GD performs worse than SGD with a moderate learning rate.

**Paired t-test for Figure 3** We also conduct statistical test to show SGD with moderate learning rate is significantly better than the other baselines. Recall that the experiments are repeated for 10 runs, at each run, we first fix a random seed, then run the four algorithms (GD/SGD with small/moderate learning rate) under the same seed. In Table 1, we report the complete results from the 10 runs. By running a paired t-test at the $5\%$ significance level, we find that SGD with moderate learning rate is significantly better than GD with moderate learning rate ($p$-value $= 0.0043$). Similarly, SGD with moderate learning rate is significantly better than SGD with small learning rate ($p$-value $= 0.012$), and is also significantly better than GD with small learning rate ($p$-value $= 0.0095$).

