# OpenReview forum: "Direction Matters: On the Implicit Bias of Stochastic Gradient Descent with Moderate Learning Rate"
_ICLR.cc/2021/Conference — ICLR 2021 Poster_

### Official Review · AnonReviewer3 · 2020-10-26
**An interesting contibution that needs some clarifications**

**Rating:** 7
**Confidence:** 3

**Review:**

In the paper the authors analyzed the convergence dynamics toward a minimum of gradient descent (GD) and stochastic gradient descent (SGD). The algorithms are considered to start in the basin of attraction of a minimum, and using discrete steps they approach the bottom. The main result of the paper concerns the fact that, with moderate learning rate, SGD approaches the minimum along the steepest direction, contrary to GD.
The authors conclude that the implicit bias of SGD does not appear in small learning rate formulation, but it becomes manifest when a moderate learning rate is considered.

* It seems from the analysis that SGD differs from GD because its learning rate is modified by the batch size and therefor it uses an "effective learning rate". The question is : what would happen if GD is taken with a learning rate that matches the effective learning rate of SGD? This situation is very similar to the one observed in [Nakkiran (2020)] (and also close to [Lewkowycz et al. (2020)]), where they consider GD and draw similar conclusions.

* Another questions is whether it is really necessary to have discrete steps to have a difference between GD and SGD in the continuous formulations. In the introduction the authors make a comparison between the implicit biases in the literature for GD and SGD claiming that they are the same because of the small learning rate. However this is not clear to me, could the authors cite papers where this is explicitly shown?
In particular there is a literature on the effect of SGD in the continuous formulation [e.g Jastrzebski et al. 2018 (arXiv:1711.04623); Zhu et al. 2019 (ICML2019); Xie et al. 2020 (arXiv:2002.03495)] using stochastic differential equations it was possible to show that SGD has an implicit temperature that adapts to the geometry of the landscape favoring, in particular, flatter solutions. In few words, do we need to rely on the discrete formulation to understand the advantages of SGD?

* The result by [Nakkiran (2020)] seems to heavily rely on having a data distribution that is highly homogeneous to observe the difference between the training and the generalization loss. Do we have evidence that this is relevant for practical situations?
I would happy have more numerical experiments as the one in Fig.2 . In particular showing the effect of increasing the learning rate in GD (and SGD) from the current "small" value to a value large enough so that it does not converge anymore. Both in the synthetic and real dataset.

The questions above block me from giving a higher rating, but I will be happy to increase the rate given satisfactory answers.

---

> ### Author Response · Authors · 2020-11-17
> **Reply to R#3**
>
> Thank you for recognizing our contributions. Regarding your concerns, we address them in the following:
>
> 1. “What would happen if GD is taken with a learning rate that matches the effective learning rate of SGD? This situation is very similar to the one observed in [Nakkiran (2020)] (and also close to [Lewkowycz et al. (2020)]), where they consider GD and draw similar conclusions.”
> - We thank you (and R#4) for raising this question. We argue that Theorem 2 precludes GD from having a legitimate large learning rate (LR) to converge along large eigenvalue directions. In specific, Eq. (6) gives the range of LR where GD falls into the regime of Theorem 2, i.e., converging along small eigenvalue directions. Note that the upper bound in Eq. (6), $n / (2\lambda_1)$, is linear in the number of training data $n$, which is very huge in practice (e.g., $n > 10^4$ for MNIST and CIFAR10). Therefore Eq. (6) already covers the largest possible LR that one can use in practice. We have updated remark 4 to include the above discussions.
>
> - Regarding the similarity with [Nakkiran, 2020], we would like to highlight the difference between [Nakkiran, 2020] and our work. Specifically, [Nakkiran, 2020] showed the separation between the test error of GD with “large” and annealing learning rate, and test error of GD with small learning rate. However, the “large” learning rate for GD in their analysis is linear in the training sample size and is impractically large as we have discussed above. In contrast, we showed that under the practically used moderate learning rate, there is a separation between the test error of SGD and the test error of GD. Therefore, our result is different from that in [Nakkiran, 2020], and our contribution is unique. We have revised remark 6 to emphasize this point.
>
> 2. “Another question is whether it is really necessary to have discrete steps to have a difference between GD and SGD in the continuous formulations.”
> - Yes, it is necessary to consider discrete steps to explain the difference between the implicit bias of GD and SGD. Regarding the reference which shows the implicit biases of GD and SGD are the same for infinitesimal learning rate, please refer to Theorem 2.1 in [Kushner & Yin, 2003]. It directly shows that SGD converges to the same ODE given by GD when the learning rate tends to zero (under mild assumptions). Therefore, in the continuous time scenario, SGD cannot be differentiated from GD.
> Regarding the references [Jastrzebski et al. 2018; Zhu et al. 2019; Xie et al. 2020] you mentioned and also cited in our paper, they use SDE to approximate SGD and cannot deal with infinitesimal step size (note the Brownian motion term in the SDE is scaled by $\sqrt{\eta}$, thus an infinitesimal step size will make it vanish and the SDE will degenerate to an ODE). Therefore, their results are orthogonal to our results and not directly comparable to ours.
>
> 3. “Do we have evidence that the data is homogeneous for practical situations, as assumed by [Nakkiran (2020)]? I would be happy to have more numerical experiments as the one in (original) Fig.2 . In particular showing the effect of increasing the learning rate in GD (and SGD) from the current "small" value to a value large enough so that it does not converge anymore. Both in the synthetic and real dataset.”
> - The distribution of real-world dataset is too complicated to be classified as “homogeneous” or “inhomogeneous”. But we do think data preprocessing (e.g., whitening and normalization) can make the data distribution more “homogeneous”. This is an interesting problem and deserves more study in the future work.
>
> - Thank you for your suggestion on the experiments. We have done additional experiments on neural networks to show the effect of increasing the learning rate in GD and SGD in Appendix D.3, Figure 4. The additional experiment results also corroborate our theory.

---

> > ### Comment · AnonReviewer3 · 2020-11-23
> > **Thank you for the reply**
> >
> > 1. 3. Thank you for the clarifications.
> >
> > 2. I do not see why the limit $\eta\rightarrow0$ should lead to the same solution as not having the noisy term. If we reduce to the classical results on Langevin dynamics and GD dynamics, we know from standard results on the Fokker-Planck equations that we obtain different dynamics leading to different asymptotic distributions. Which means that in the particular case of a covariance matrix given by the identity we obtain different dynamics in the limit. This is why I do not see why things should change having a more general covariance matrix.

---

> > > ### Author Response · Authors · 2020-11-24
> > > **Clarification on the second point**
> > >
> > > We are glad to hear that your questions/concerns 1 and 3 have been addressed.
> > >
> > > As for the second point, we agree that gradient Langevin dynamics (GLD) is different from GD with infinitesimal learning rate, yet we emphasize that GLD is also different from SGD with infinitesimal learning rate, since they have different scalings in the noise term. To see this, let us make a direct comparison between the vanilla SGD and GLD in both discrete and continuous cases, as summarized as follows in our notations. For a rigorous treatment, please see [1,2] and references thereafter.
> > >
> > > [Discrete SGD] $w_{t+1} = w_{t} - \eta \nabla L(w_t) + {\color{red}\eta} \Sigma_t^{\frac{1}{2}} \xi_t $
> > >
> > > [Discrete GLD] $w_{t+1} = w_t - \eta \nabla L(w_t) + {\color{red}\sqrt{\eta}} \Sigma_t^{\frac{1}{2}} \xi_t $
> > >
> > > where $\eta$ is the learning rate, $\Sigma_t$ is the diffusion matrix (i.e., covariance matrix of the gradient noise), and $\xi_t$ is a martingale difference sequence with zero mean and identity covariance.
> > >
> > > [Continuous SGD] $d w_t = -\nabla L(w_t) d t + {\color{red}\sqrt{\eta}} \Sigma_t^{\frac{1}{2}} d B_t$
> > >
> > > [Continuous GLD] $d w_t = -\nabla L(w_t) d t + \Sigma_t^{\frac{1}{2}} d B_t$
> > >
> > > where $B_t$ is Brownian motion.
> > >
> > > From the above equations, we can see that
> > > - SGD does not reduce to the classical gradient Langevin dynamics when $\eta$ goes to zero. The noise term in [Continuous SGD] scales with $\sqrt{\eta}$, in contrast to the noise term in [Continuous GLD] that is independent of $\eta$. Thus as $\eta$ tends to zero, the Fokker-Planck equation for [Continuous SGD] gives a Dirac distribution, but the Fokker-Planck equation for  [Continuous GLD] gives a non-trivial distribution in general.
> > > - Instead, SGD reduces to GD/gradient flow as $\eta$ tends to zero, which is proved by Theorem 2.1 in [Kushner & Yin, 2003].
> > >
> > > [1] Dalalyan, Arnak S. "Theoretical guarantees for approximate sampling from smooth and log-concave densities." arXiv preprint arXiv:1412.7392 (2014).
> > >
> > > [2] Li, Qianxiao, Cheng Tai, and E. Weinan. "Stochastic modified equations and adaptive stochastic gradient algorithms." International Conference on Machine Learning. PMLR, 2017.

---

> > > > ### Comment · AnonReviewer3 · 2020-11-24
> > > > **Thank you for the reply**
> > > >
> > > > Thank you for addressing my concern. I was indeed considering the wrong scaling.

---

### Official Review · AnonReviewer1 · 2020-10-27
**Interesting and novel analysis but I have a few concerns**

**Rating:** 6
**Confidence:** 3

**Review:**

Summary:
In this paper, an implicit bias of SGD and GD in terms of the direction of convergence points is studied. This study shows that, in a setting of linear regression, SGD and GD converge to different directions, which are determined by the largest/smallest eigenvectors of a data matrix when the learning rate is moderately large. Experiments using synthetic data and Fashion MNIST support the theoretical results.


Detailed comments:
First of all, I don't have much experience in the analysis of SGD/GD and my assessment for technical points may miss some important points.

Overall, the paper is well written. The motivation and problem setting are clearly written. Related work is sufficiently introduced.

The main theoretical results (Theorems 1, 2, 4) are interesting. As far as I know, there's no study to reveal the implicit bias in terms of direction. However, I have several concerns.

1. In the data generation process, the noiseless output y = <w*, x> is assumed. However, the output often contains observations noise such as an additive Gaussian model as y = <w*, x> + $\xi$ where $\xi$ is small Gaussian noise. Would it be possible to show similar results in the noisy case?

2. More importantly, the benefit of the directional bias is not clear. Theorem 4 shows SGD achieves $\epsilon$-optimal solution and GD achieves $M$-suboptimal solution. However, we cannot conclude that SGD solution is better than GD solution because the generalization performance depends on the unknown constants $\epsilon$ and $M$. So, GD may win in some cases but SGD may win in other cases, but it seems there is no way to know in what conditions $\epsilon$ and $M$ satisfy some specific values so that SGD beats GD. Would it be possible to clarify the conditions?

---

> ### Author Response · Authors · 2020-11-17
> **Reply to R#1**
>
> Thank you for appreciating the novelty and writing of our work. We answer your concerns as follows:
>
> 1. “Noisy labels”.
> - Our main focus in this work is to justify the directional bias for SGD with moderate learning rate. Therefore our results are presented for a realizable (interpolating) overparameterized linear model to keep the statements concise. Nevertheless, our results are applicable to noisy label settings as well. Note that given a sufficient overparameterization, a linear model can fit the noisy labels perfectly (in other words, interpolating the training data points with noisy labels) with high probability [Bartlett et al., 2020]. Then we only need to revise our current results by taking a conditional expectation over the i.i.d. zero mean label noise.
>
> 2. “Constants in Theorem 4”.
> - We apologize for the imprecise descriptions on the constants $\epsilon$ and $M$ in the original Theorem 4. In fact, according to the formal version of Theorem 4 in Appendix B, we have $\epsilon = o(1)$ and $M = \gamma_1 / \gamma_n - o(1)$, where $\gamma_1$ and $\gamma_n$ are the largest and smallest (non-zero) eigenvalues of the data covariance matrix, respectively. Since $M - (1+\epsilon) > \gamma_1/\gamma_n - 1 - o(1) > 0$, we have a separation between the test error of SGD with moderate learning rate and the test error of GD/SGD with small learning rate. We have made the constants in Theorem 4 precise in the revision. We also add a discussion to emphasize this separation after Theorem 4 in the revision.

---

> > ### Comment · AnonReviewer1 · 2020-11-24
> > **Re**
> >
> > Sorry for my late reply (I was busy with AAAI review) and thank you for your response.
> >
> > > “Noisy labels”
> >
> > I'm still puzzled. Statistically speaking, it's common to analyze the generalization error of a method under some observation noise (in this case, the nose on $y$). However, your comment "a linear model can fit the noisy labels perfectly" seems to mean that you only consider the case that the model is overfitting to the training data and the generalization error is not minimized. Also, your suggestion "Then we only need to revise our current results by taking a conditional expectation over the i.i.d. zero mean label noise" sounds untrivial for me. Can you say something like Theorem 4 for the noisy case?
> >
> > > “Constants in Theorem 4”
> >
> > Thank you for your revision, which clarifies my concern.

---

> > > ### Author Response · Authors · 2020-11-24
> > > **Clarification on the label noise**
> > >
> > > Thank you for your reply. We are glad that the revised statement of Theorem 4 is clear to you. In the following, we explain how our results can be extended to the case where the training data label contains i.i.d. additive noise. Let us denote the noisy label by $y = z + e$, where $z = w _* ^\top x$ is the clean label, and $e$ is the i.i.d. zero mean label noise with variance $\sigma ^2$. We also define $X = (x_1, \dots, x_n)$, $Y = (y_1, \dots, y_n)^\top$, $Z = (z_1, \dots, z_n)^\top$ and $E = (e_1, \dots, e_n)^\top$. Recall that $P$ is the projection operator onto the column space of $X$.
> > >
> > > We first argue that in the noisy case, Theorems 1-3 still hold. Note that the overparameterized linear model can interpolate the noisy label [Belkin et al., 2019, Bartlett et al., 2020]. Therefore, if we replace the clean label with a noisy label, our proofs still hold.
> > >
> > > As for Theorem 4, recall that the estimation error is $\\|P(w ^{alg} - w _\*)\\| _2 ^2$.
> > > In the noiseless case, the minimum-norm solution is $P(w _*) = X(X ^\top X) ^{-1}Z$; however in the noisy case, the minimum-norm solution becomes $\hat{w} = X(X ^\top X) ^{-1}Y$, which is not $P(w _*)$ thanks to the label noise. To deal with the label noise, we consider the expected estimation error conditional on the label noise (denoted by $\mathbb{E} _{Y|X}$):
> > > $$
> > > \begin{aligned}
> > > \mathbb{E} _{Y|X}[\\|P(w ^{alg} - w _\*)\\|_2^2] &= \mathbb{E} _{Y|X} [\\|P(w ^{alg} - \hat{w}) + P(\hat{w} - w _\*) \\| _2 ^2]  \\\\
> > > &= \mathbb{E} _{Y|X}[\\|P(w ^{alg} - \hat{w}) \\| _2 ^2] + 2\mathbb{E} _{Y|X}[\langle P(w ^{alg} - \hat{w}), P(\hat{w} - w _\*) \rangle] +\mathbb{E} _{Y|X} [P(\hat{w} - w _\*) \\| ^2 _2] \\\\
> > > &= \mathbb{E} _{Y|X}[\\|P(w ^{alg} - \hat{w}) \\| _2 ^2] + 2\mathbb{E} _{Y|X}[\langle P(w ^{alg} - \hat{w}), P(\hat{w} - w _\*) \rangle] + \sigma ^2 Tr((X ^\top X)^{-1}),
> > > \end{aligned}
> > > $$
> > > where the last equality is because $\mathbb{E} _{Y|X} [\\|P(\hat{w} - w _\*) \\| _2 ^2] = \mathbb{E} _{Y|X} [\\| X(X ^\top X)^{-1}E \\| _2 ^2]=\mathbb{E} _{Y|X} [Tr(X(X ^\top X)^{-1}EE ^\top (X ^\top X) ^{-1} X ^\top)]=
> > > \sigma^2 Tr((X ^\top X) ^{-1})$.
> > >
> > > Consider the RHS of the decomposition, we have:
> > > - The first term is identical to the estimation error in the noiseless setting, and we can directly apply the results in the current Theorem 4.
> > > - The second term is approximately zero due to the following two facts:
> > >     * For $P(w ^{alg} - \hat {w})$, (1) based on Theorems 1-3, its direction is nearly parallel to the largest  (resp. smallest) eigenvalue direction of the data matrix $X X ^\top$ for SGD with moderate learning rate (resp. GD), which is independent of the label noise $E$; and (2) its magnitude is also independent of the label noise since we fix an $\alpha$-level set, i.e., $\frac{1}{n}\\|(w ^{alg} - \hat {w}) ^\top X X ^\top (w ^{alg} - \hat {w}) \\|_2^2 = \alpha$.
> > >     * For $P(\hat{w} - w _\*)$, we have $\mathbb{E} _{Y|X}[P(\hat{w} - w _\*)] = \mathbb{E} _{Y|X}[X(X ^\top X) ^{-1}Y-X(X ^\top X) ^{-1}Z]=0$.
> > >
> > > - The third term is independent of the algorithm/hyperparameters, and we can ignore it when comparing the generalization ability of different algorithms.
> > >
> > > As a sanity check, when there is no noise (i.e., $\sigma ^2 = 0$), we have $P(\hat{w}) = P(w _\*)$, thus the second and third terms are both zero. This immediately reduces to the results in the current Theorem 4.
> > >
> > > Based on the above reasoning, we argue that all Theorems 1-4 can be extended to the noisy label case for justifying the directional bias and its effect on algorithm generalization. We are happy to discuss more should there be further unclearness.

---

### Official Review · AnonReviewer4 · 2020-10-28
**Too specific with weak experiments.**

**Rating:** 6
**Confidence:** 3

**Review:**

This paper analyzes the differences in the convergence of SGD and GD when using "small" and "moderate" learning rates to shed light on why SGD with "moderate" and annealing learning rates perform well in practice. Focusing on an overparametrized linear regression problem, the paper claims that SGD with a "moderate" learning converges differently (along the large eigenvalue directions) compared to SGD/GD with a "small" learning rate (along the small eigenvalue directions).  They further show analytically for this problem that there exists a learning rate schedule (with moderate learning rates) such that SGD will perform + generalize well, while there exist small learning rate schedules such that GD will perform + generalize poorly.

This paper is clearly written and presents a clear illustration of their hypothesis for why SGD with a moderate learning rate performs well in practice through Figure 1 and a simple linear regression example in section 3. The paper focuses most of its analysis on a toy example problem (overparametrized linear regression) and the empirical experiments do not clearly support the paper's claims.

This paper should be rejected as is. Although the motivating example and illustrations are interesting, the analysis for the overparameterized linear regression problem is too specific to clearly claim that this "directional bias" is why SGD with moderate learning rates is successful.

I have two main concerns: (1) in the special case of overparametrized linear regression, Theorem 2 only shows that GD with small learning rates converge along the smallest eigenvalue direction. It does not preclude GD from having the same behavior as SGD for larger learning rates. I also do not agree with remark 4.
(2) it's not clear how or why the results for overparameterized linear regression would extend to generic NN losses. The example experiments (Fig 2) need more detail to be clear and are not sufficient to support the hypothesis in the paper.

Questions:
In section 3, doesn't Eq (3) show that GD will behave identically to SGD if the learning rate is doubled? What is considered a "moderate" learning rate for GD + SGD does not need to be identical, right?

For the overparametrized regression, why do we not include some form of regularization on the weights (as would commonly be done in practice)? Does this affect the results?

What is the minibatch size for SGD in the experiments? How are the results (in Figure 2a + 2b) sensitive to random initialization?

Minor comment: Page 6: the definition of level set probably needs a $\leq$ sign.

---

> ### Author Response · Authors · 2020-11-17
> **Reply to R#4**
>
> Thanks for recognizing the novelty and writing of our work. Regarding your concerns, please see below:
>
> 1. “Theorem 2 only shows that GD with small learning rates converge along the smallest eigenvalue direction. It does not preclude GD from having the same behavior as SGD for larger learning rates.”
> - This is a huge misunderstanding. We apologize that we did not make it clear. In fact, Theorem 2 indeed precludes GD from having a large learning rate (LR) to converge along large eigenvalue directions. In specific, Eq. (6) gives the range of LR where GD falls into the government of Theorem 2, i.e., converging along small eigenvalue directions. Note that the upper bound in Eq. (6), $n / (2\lambda_1)$, is linear in the training sample size, which is very large for typical datasets (e.g., $n > 10^4$ for MNIST and CIFAR10). Therefore, Eq. (6) has already covered the largest possible LR that one can use in practice.
>
> 2. “I also do not agree with remark 4.”
> - We thank R#4 to point out the issue on remark 4. We have updated remark 4 to include the above discussions on Theorem 2. We hope the new remark is crystal clear that it is impossible to have a numerically stable LR such that GD converges along large eigenvalue directions.
>
> 3. “It's not clear how or why the results for overparameterized linear regression would extend to generic NN losses.”
> - We thank the reviewer for this comment. We agree with the reviewer that proving directional bias of SGD with moderate learning rate for neural networks is still an open problem. However, even in the linear regression setting, there is no such kind of result before our work. Without a full characterization of directional bias of SGD for the simplest possible problem — the linear regression — it seems unlikely that one can prove it for neural networks. As our work is the first work on the directional bias of SGD, we believe we have taken an important step along this direction and the contribution of our work is very significant.
>
> - In our revision, we have added new experiments (Figure 2(b)) on NN to support our theory, which also shed light on understanding the directional bias for SGD for general NNs. A brief description of the experiments is in the “reply to all reviewers” response.
>
> 4. “The example experiments (original Fig 2) need more detail to be clear and are not sufficient to support the hypothesis in the paper.”
> - We add a new section (Appendix D) to describe the full details about our experiments. We have also provided the Code in the supplemental material for the reproducibility of our experiments. We argue that Figure 2(a) does verify our theory in an overparameterized linear model. As for NNs, new experiments are provided in (new) Figure 2(b) to support our theory. Should there be any missed detail, please let us know and we are happy to clarify.
>
> 5. “In section 3, doesn't Eq (3) show that GD will behave identically to SGD if the learning rate is doubled? What is considered a "moderate" learning rate for GD + SGD does not need to be identical, right?”
> - Your comment is absolutely correct for this toy example, which involves only TWO training data points and is meant for proof of concept. In general, however, according to Theorem 2 and our previous discussions, the “effective” LR for GD scales linearly in the training sample size $n$, which is quite huge in practice. So doubling the learning rate of GD cannot make GD and SGD (with moderate LR) behave the same.
>
> 6. “For the overparameterized regression, why do we not include some form of regularization on the weights (as would commonly be done in practice)? Does this affect the results?”
> - As our work is the first to study the implicit bias (i.e., implicit regularization) of SGD with moderate learning rate, we chose to exclude explicit regularizers. By adding explicit regularization, we think our implicit bias result will be affected in general, and it is an interesting future work to study the interplays between explicit regularization and implicit regularization.
>
> 7. “What is the minibatch size for SGD in the experiments? How are the results (in Figure 2a + 2b, original) sensitive to random initialization?”
> - (1) The minibatch size of SGD is 25. Full experiment details are in Appendix D. (2) Results are robust to random initialization. In the revision, we have updated experiments for neural networks to include the standard deviation computed over 10 runs, which shows that the results are quite robust. We have also attached Code for reproducibility.
>
> 8. “Page 6: the definition of level set probably needs a <= sign.”
> - This is a misunderstanding. Level sets are indeed defined with “=”. Those defined with “<=” are called sublevel sets.
>
> We hope your concerns have been addressed, and are looking forward to your further comments.

---

> > ### Comment · AnonReviewer4 · 2020-11-19
> > **Clarification**
> >
> > The authors comments and revisions have cleared my misunderstandings of this paper. The new experiments on FashionMNIST with convnets is very welcome for helping connect the results to practice.
> >
> > * 1 Theorem 2 still does not (technically) prevent learning rates greater than $n/2\lambda_1$ from behaving similarly to SGD (for moderate $n$); however the authors are correct that for large enough $n$ the range of these learning rates will *eventually* cover all reasonable learning rate schemes. I'm still curious of when the learning rates outside of Theorem 2 for GD become unstable, to make this more rigorous.
> > * 2 I apologize for my confusion on Remark 4. For large $n$, I now agree with the authors.
> > * 3/6 My concern was the reduction to overparametrized linear regression without regularization, might prevent the novel insights to the differences between SGD and GD from applying to the training of general NN models in practice. Or might ignore other possible sources of improvement (e.g. stochasticity in non-convex optimization).
> > * 4/7 Thank you for adding Appendix D to clarify the experiments. Are the NN results robust? The standard deviation ranges in Figure 3 for moderate LR GD and moderate SGD appear to overlap. In particular, moderate LR GD has a very large standard deviation.
> > * 8 You are correct. My mistake.

---

> > > ### Author Response · Authors · 2020-11-20
> > > **Thanks for your reply**
> > >
> > > We appreciate your prompt reply, and are glad to see many of your concerns have been addressed. Regarding the new comments, we make the following clarifications:
> > >
> > > 1. “ Theorem 2 still does not (technically) prevent learning rates greater than $n/2\lambda_1$ from behaving similarly to SGD (for moderate n) ”
> > > - We agree with you that Theorem 2 does not prevent GD from behaving similarly to SGD with moderate LR by having a very large LR. Theorem 2 can be complemented by the following result on large LR for GD:
> > > [large LR regime] If the initial LR is larger than $n/\lambda_1$, then by Theorem 1 (setting $b = n$), provided a proper LR annealing, GD can still converge along the large eigenvalue directions.
> > >
> > > 2. “I'm still curious about when the learning rates outside of Theorem 2 for GD become unstable, to make this more rigorous.”
> > > - In our setting, if the LR is larger than $n / \gamma_n$, GD diverges along any direction; if the LR is larger than $n / \gamma_1$, GD is convergent along some directions, but it requires learning rate annealing to converge to the minimum; if the LR is smaller than $n / \gamma_1$, GD converges to the minimum without requiring LR annealing.
> > >
> > > 3. “My concern was the reduction to overparameterized linear regression without regularization, might prevent the novel insights to the differences between SGD and GD from applying to the training of general NN models in practice. Or might ignore other possible sources of improvement (e.g. stochasticity in non-convex optimization).”
> > > - We would like to emphasize that the focus of this paper is to prove the directional bias of SGD in overparameterized linear regression, which is a new type of implicit bias that has not been discovered in previous studies. Extension of our theory to general NN models is beyond the focus of our paper, and we only have some empirical results to demonstrate it. We appreciate your suggestion, and we have listed the extension of our results to general NN models as an important future work in the revision (see Section 7).
> > >
> > > 4. “Are the NN results robust? The standard deviation ranges in Figure 3 for moderate LR GD and moderate SGD appear to overlap. In particular, moderate LR GD has a very large standard deviation.”
> > > - Yes, the NN results are actually robust. We have provided (new) Table 1 in Appendix D.3 to report the complete results of the four algorithms from 10 runs. After doing a paired t-test (at the 5% significance level), we find SGD with moderate LR outperforms GD with moderate LR with p-value equals to $0.0043$, which suggests that the result is statistically significant. Please refer to the paragraph “Paired t-test for Figure 3” in Appendix D.3 for the complete statistical tests.

---

### Author Response · Authors · 2020-11-17
**Reply to all Reviewers: Revised PDF, Important New Experiments**

We thank all reviewers for their constructive comments.
- We have revised the paper accordingly to address the concerns raised. Newly added contents are marked in the magenta color.
- We have also conducted new experiments in neural networks for supporting our theory. Please see Figure 2(b) and Appendix D.3 for details. We briefly describe the empirical results as follows:
We consider a convolutional neural network on a subset of the FashionMNIST dataset. And we empirically compare the convergence directions of SGD and GD in different learning rate regimes. Note that the neural network is non-convex and has multiple minima. Thus in order to measure the convergence direction, we calculate the “relative Rayleigh quotient”, which is defined as the Rayleigh quotient of the convergence direction divided by the maximum absolute eigenvalue of the Hessian (at the same point). According to the new experimental result in Figure 2(b), it is evident that SGD with moderate learning rate converges along relatively large eigenvalue directions while GD/SGD with small learning rate converges along relatively small eigenvalue directions. This result supports our theory in neural networks.

---

### Decision · Program_Chairs · 2021-01-10
**Final Decision**

**Decision:**

Accept (Poster)

**Comment:**

This work compares and contrasts the learning rate dynamics of GD and SGD and shows that under practical learning rate settings, SGD is biased to approach the minimum along the direction of steepest descent, leading to better performance. Reviewers agree that the theoretical results are significant. The authors satisfactorily responded to reviewers’ questions and improved the paper’s clarity during the discussion phase.